# DASH: Warm-Starting Neural Network Training Without Loss of Plasticity Under Stationarity

Baekrok Shin [* 1]  Junsoo Oh [* 1]  Hanseul Cho [1]  Chulhee Yun [1]

## Abstract

Warm-starting neural networks by initializing them with previously learned weights is appealing, as practical neural networks are often deployed under a continuous influx of new data. However, it often leads to *loss of plasticity*, where the network loses its ability to learn new information, resulting in worse generalization compared to training from scratch. This occurs even under stationary data distributions, and its underlying mechanism is poorly understood. We develop a framework emulating real-world neural network training and identify noise memorization as the primary cause of plasticity loss when warm-starting on stationary data. Motivated by this, we propose **Direction-Aware SHrinking (DASH)**, a method aiming to mitigate plasticity loss by selectively forgetting memorized noise while preserving learned features. We validate our approach on vision tasks, demonstrating improvements in test accuracy and training efficiency.

## 1. Introduction

When training a neural network on a gradually changing dataset, the model tends to lose its *plasticity*, which refers to the model's ability to adapt to new information (Lyle et al., 2023b; Dohare et al., 2021; Nikishin et al., 2022). This phenomenon is particularly relevant in scenarios with non-stationary data distributions, such as reinforcement learning (Igl et al., 2020; Nikishin et al., 2022) and continual learning (Wu et al., 2021; Chen et al., 2023; Kumar et al., 2023). A common explanation is that previously learned information becomes less relevant over time, necessitating models to overwrite the existing knowledge (Lyle et al., 2023b). Under this viewpoint, various efforts have been made to mit-

igate the loss of plasticity, such as resetting layers (Nikishin et al., 2022), regularizing weights (Kumar et al., 2023), and modifying architectures (Nikishin et al., 2023; Dohare et al., 2021; Sokar et al., 2023; Lyle et al., 2023a; Lee et al., 2023).

Perhaps surprisingly, a similar phenomenon occurs in supervised learning settings, even where new data points sampled from a stationary data distribution keep being introduced to the dataset during training. It is counterintuitive, as one would expect advantages in both generalization performance and computational efficiency when we *warm-start* from a model pre-trained on data points of the same distribution. For a particular example, when a model is pre-trained using a portion of a dataset and then we resume the training with the whole dataset, the generalization performance is often worse than a model trained from scratch (i.e., *cold-start*), despite achieving similar training accuracy (Ash & Adams, 2020; Berariu et al., 2021; Igl et al., 2020). Liu et al. (2020) report a similar observation: training neural networks with random labels leads to a spurious local minimum which is challenging to escape from, even when retraining with a correctly labeled dataset. Interestingly, Igl et al. (2020) found that pre-training with random labels followed by the corrected dataset yields better generalization performance than pre-training with a small portion of the (correctly labeled) dataset and then training with the full, unaltered dataset. It is striking that warm-starting leads to such a severe loss of performance, even worse than that of a cold-started model or a restarted model from a pre-trained one with random labels, despite the stationarity of the data distribution.

These counterintuitive results prompt us to investigate the underlying reasons for them. While some studies have attempted to explain the loss of plasticity in deep neural networks (DNNs) under non-stationarity (Lyle et al., 2023b; Sokar et al., 2023; Lewandowski et al., 2023), their empirical explanations rely on various factors, such as model architecture, datasets, and other variables, making it difficult to generalize the findings (Lewandowski et al., 2023; Lyle et al., 2023a). Moreover, there is limited research that explores why warm-starting is problematic in stationary settings, highlighting the lack of a fundamental understanding of the loss of plasticity phenomenon in both stationary and non-stationary data distributions.

[*]Equal contribution  [1]Kim Jaechul Graduate School of AI, KAIST. Correspondence to: Chulhee Yun <chulhee.yun@kaist.ac.kr>.

Accepted to the Workshop on Advancing Neural Network Training at International Conference on Machine Learning (WANT@ICML 2024).

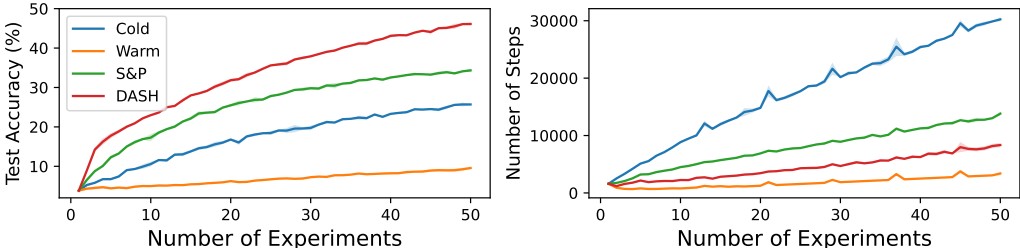

*Figure 1.* Performance comparison of various methods on Tiny-ImageNet using ResNet-18. The same hyperparameters are used across all methods. In each experiment, a constant number of data points are added to the existing training dataset. Models are trained until achieving 99.9% train accuracy before proceeding to the next experiment; the plot on the right reports the number of update steps executed in each experiment. Results are averaged over three random seeds. "Cold" refers to cold-starting and "Warm" refers to warm-starting. The Shrink & Perturb (S&P) method involves shrinking the model weights by a constant factor and adding noise (Ash & Adams, 2020). Notably, DASH, our proposed method, achieves better generalization performance compared to both training from scratch and S&P, while requiring fewer steps to converge.

### 1.1. Our Contributions

In this work, we aim to explain why warm-starting leads to worse generalization compared to cold-starting, focusing on the stationary case. We propose an abstract framework that combines the popular feature learning framework initiated by Allen-Zhu & Li (2020) with a recent approach by Jiang et al. (2024) that studies feature learning in a combinatorial and abstract manner. Our analysis suggests that warm-starting leads to overfitting by memorizing noise present in the newly introduced data rather than learning new features.

Inspired by this finding, we propose Direction-Aware SHrinking (DASH), which aims to encourage the model to forget memorized noise without affecting previously learned features. This enables the model to learn features that cannot be acquired through warm-starting alone, enhancing the model's generalization ability. We validate DASH using the online learning setting from Ash & Adams (2020), employing various models, datasets, and optimizers. As an example, Figure 1 shows the promising results in terms of both test accuracy and training time.

### 1.2. Related Works

**Loss of Plasticity.** Research has aimed to understand and mitigate loss of plasticity in non-stationary data distributions. Lewandowski et al. (2023) explain that loss of plasticity co-occurs with a reduction in the Hessian rank of the training objective, while Sokar et al. (2023) attribute it to an increasing number of inactive neurons during training. Lyle et al. (2023b) find that changes in the loss landscape curvature caused by non-stationarity lead to loss of plasticity. Methods addressing this issue in non-stationary settings include recycling dormant neurons (Sokar et al., 2023), regularizing weights towards initial values (Kumar et al., 2023), and combining techniques (Lee et al., 2023) like layer normalization (Ba et al., 2016), Sharpness-Aware Minimization (SAM)

(Foret et al., 2020), resetting layers (Nikishin et al., 2022), and Concatenated ReLU activation (Shang et al., 2016).

However, these explanations and methods diverge from the behavior observed in stationary data distributions. Techniques aimed at mitigating loss of plasticity under non-stationarity are ineffective under stationary distributions, as shown in Appendix A.1, in line with the observations in Lee et al. (2023). While some works study the warm-starting problem in stationary settings, they rely on empirical observations without theoretical analysis (Ash & Adams, 2020; Berariu et al., 2021; Achille et al., 2018). The most relevant work by Ash & Adams (2020) introduces the Shrink & Perturb (S&P) method, which mitigates loss of plasticity in stationary settings to some extent by shrinking all weight vectors by a constant factor and adding noise. However, they do not explain why this phenomenon occurs or why S&P is effective. We develop and analyze a theoretical framework to explain why warm-starting suffers even under stationary data distribution. Based on our findings, we propose a method that shrinks the weight vector in a direction-aware manner to maintain properly learned features.

**Feature Learning in Neural Networks.** Recent studies have investigated how training methods and network architectures influence generalization performance, focusing on data distributions with label-dependent features and label-independent noise (Allen-Zhu & Li, 2020; Cao et al., 2022; Jelassi & Li, 2022; Zou et al., 2023; Deng et al., 2023; Oh & Yun, 2024). In particular, Shen et al. (2022) examine a data distribution consisting of varying frequencies of features and large strengths of noise, emphasizing the significance of feature frequencies in learning dynamics. Jiang et al. (2024) propose a novel feature learning framework based on their observations in real-world scenarios, which also involves features with different frequencies but considers the learning process as a discrete sampling process. Our

proposed framework extends these ideas by incorporating features with varying frequencies, noise components, and the discrete learning process while introducing a more intricate learning process that captures the key aspects of feature learning dynamics in gradually expanding datasets.

## 2. A Framework of Feature Learning

Shen et al. (2022) consider a neural network trained on data with features of different frequencies and noise components stronger than the features. The gradient of the loss for each single data point aligns more with the noise than the features due to the larger scale of noise, making the model more likely to memorize noise rather than learn features. However, an identical feature appears in many data points, while noise appears only once and do not overlap across data points. Thus, if a feature appears in sufficiently high frequency in the dataset, the model can learn the feature. Thus, the model's learning of features or noise depends on the frequency of features and the strength of noise.

Inspired by Shen et al. (2022), we propose a novel discrete feature learning framework. This section introduces a framework describing a single experiment, while Section 3 analyzes expanding datasets scenarios. As our focus is on gradually expanding datasets, carrying out the (S)GD analysis over many experiments as in Shen et al. (2022) is highly challenging. Instead, we adopt a discrete learning process similar to Jiang et al. (2024) but propose a more intricate process reflecting key ideas from Shen et al. (2022). In doing so, we generalize the concept of plasticity loss and analyze it without assuming any particular hypothesis class for a more comprehensive understanding, whereas existing works are limited to specific architectures.

### 2.1. Training Process

We consider a classification problem with $C$ classes, and data are represented as $(\boldsymbol{x}, y) \in \mathcal{X} \times [C]$, where $\mathcal{X}$ denotes the input space. A data point is associated with a combination of class-dependent features $\mathcal{V}(\boldsymbol{x}) \subset \mathcal{S}_y$ where $\mathcal{S}_c = \{v_{c,1}, v_{c,2}, \ldots, v_{c,K}\}$ is the set of all features for each class $c \in [C]$. Also, every data point contains data-specific noise which is class-independent.

The model $f : \mathcal{X} \to [C]$ sequentially learns features based on their frequency. The training process is described by the set of learned features $\mathcal{L} \subset \mathcal{S} \triangleq \bigcup_{c \in [C]} \mathcal{S}_c$ and the set of data points with non-zero gradients $\mathcal{N} \subset \mathcal{T}$, where $\mathcal{T} = \{(\boldsymbol{x}_i, y_i)\}_{i \in [m]}$ denotes a training set. The set $\mathcal{N}$, representing the data points with non-zero gradients, will be defined below. The frequency of a feature $v$ in data points belonging to $\mathcal{N}$ is denoted by

$$g(v; \mathcal{T}, \mathcal{N}) = \frac{1}{|\mathcal{T}|} \sum_{(\boldsymbol{x}, y) \in \mathcal{N}} \mathbb{1}(v \in \mathcal{V}(\boldsymbol{x})),$$

where $\mathbb{1}(\cdot)$ is the indicator function, which equals 1 if the condition inside the parentheses is true and 0 otherwise. At each step of training, if $\mathcal{L}$ and $\mathcal{N}$ are given, the model chooses the most frequent feature among the features not yet learned, i.e., arbitrarily choose $v \in \arg\max_{u \in \mathcal{S} \setminus \mathcal{L}} g(u; \mathcal{T}, \mathcal{N})$.

The model decides whether to learn a selected feature $v$ by comparing its signal strength, represented by $|\mathcal{T}| \cdot g(v; \mathcal{T}, \mathcal{N})$, with the signal strength of noise, given by $\gamma$, which reflects the key ideas of Shen et al. (2022). If the frequency of the selected feature $v$ is no less than the threshold $\gamma/|\mathcal{T}|$, i.e., $g(v; \mathcal{T}, \mathcal{N}) \geq \gamma/|\mathcal{T}|$, the model learns $v$ and adds it to its set of learned features $\mathcal{L}$. The feature learning process continues until the model reaches a point where the selected feature $v$ has $g(v; \mathcal{T}, \mathcal{N}) < \gamma/|\mathcal{T}|$, indicating that the signal strength of every remaining feature is weaker than that of noise. At this point, the feature learning process ends.

We consider a data point $\boldsymbol{x}$ to be *well-classified* if the model $f$ has learned at least $\tau$ features from $\mathcal{V}(\boldsymbol{x})$, i.e., $|\mathcal{L} \cap \mathcal{V}(\boldsymbol{x})| \geq \tau$, where $\tau < K$. In this case, we consider $\boldsymbol{x}$ to have a zero gradient, meaning it cannot further contribute to the learning process. Throughout the feature learning process, the set $\mathcal{N}$ of data points with non-zero gradients is dynamically updated as new features are learned. At each step, when the model successfully learns a new feature, we update $\mathcal{N}$ by removing the data points that satisfy $|\mathcal{L} \cap \mathcal{V}(\boldsymbol{x})| \geq \tau$, as they become well-classified due to the newly learned feature.

If the feature learning process ends and the model has learned as many features as it can, the remaining data points that have non-zero gradients will be *memorized* by fitting the random noise present in them and will be considered to have zero gradients. This step concludes the training process. A detailed algorithm of the learning process can be found in Algorithm 2 in Appendix D.

### 2.2. Discussion on Training Process

In our framework, the model selects features based on their frequency in the set of unclassified data points $\mathcal{N}$. The intuition behind this approach is that features appearing more frequently in the set of data points will have larger gradients, leading to larger updates, and we treat $g(v; \mathcal{T}, \mathcal{N})$ as a proxy of the gradient for a particular feature $v$. As a result, the model prioritizes learning these high-frequency features in a sequential manner. However, if the frequency $g(v; \mathcal{T}, \mathcal{N})$ of a particular feature $v$ is not sufficiently large, such that the total occurrence of $v$ is less than the strength of the noise, i.e., $|\mathcal{T}| \cdot g(v; \mathcal{T}, \mathcal{N}) < \gamma$, the model may struggle to learn that feature. Consequently, the model will prioritize learning the noise over the informative features. When this situation arises, the learning procedure becomes sub-optimal because

the model fails to capture the true underlying features of the data and instead memorizes the noise.

The threshold $\tau$ determines when a data point is considered well-classified and acts as a proxy for the dataset's complexity. A higher $\tau$ requires the model to learn more features for correct predictions, while a lower $\tau$ allows accurate predictions with fewer learned features. Experiments in Appendix B Figure 13 and 14 support this interpretation.

### 2.3. Prediction Process and Training Time

The model predicts on unseen data points by comparing the learned features with features present in a given data point $x$. If the overlap between the learned feature set $\mathcal{L}$ and the features in $x$, denoted as $\mathcal{V}(x)$, is at least $\tau$, i.e., $|\mathcal{V}(x) \cap \mathcal{L}| \geq \tau$, the model correctly classifies the data point. Otherwise, the model resorts to random guessing.

Accurately measuring training time within our discrete learning framework is challenging. To address this, we introduce an alternative for training time based on empirical observations. We consider the number of training data points with non-zero gradients at the initial stage of training as a proxy for training time, which represents the amount of "learning" required for the model to classify all data points correctly. To verify this, we conducted experiments with CIFAR-10 on ResNet-18. We used the gradient norm as a proxy for the number of data points with non-zero gradients and investigated its correlation with the number of training steps required to achieve 99.9% training accuracy. Figures 10 and 11 in Appendix B show that larger initial gradient norms correlate with longer convergence times in real-world neural network training. Therefore, we consider $|\mathcal{N}|$ at the initial state of training as a proxy for the training time until the model perfectly fits the training data, which represents the number of training data points that need to be newly well-classified throughout the training procedure.

*Remark* 2.1. Nakkiran et al. (2021) also observe that in real-world neural network training, when other components are fixed, the training time increases with the number of data points to learn.

## 3. Warm-Starting vs. Cold-Starting

### 3.1. Experiments with Expanding Dataset

In this section, we establish the setup where the dataset grows after each $j$-th experiment, with $j \in \mathbb{N}$ to compare warm-start and cold-start within our proposed learning framework. To better understand the loss of plasticity under stationary data distribution, we consider an extreme form of stationarity where the frequency of each feature combination remains constant in each chunk. We investigate if the loss of plasticity can manifest even under this strong stationarity. The detailed description of the dataset across

the entire experiment is as follows:

**Assumption 3.1.** In each $j$-th experiment, we are provided with a training dataset $\mathcal{T}_j := \{(x_{i,j}, y_{i,j})\}_{i \in [n]}$ with $n$ samples. For each class $c \in [C]$ and each possible feature combination $\mathcal{A} \subset \mathcal{S}_c$, we assume that $\mathcal{T}_j$ contains exactly $n_{\mathcal{A}} \geq 1$ data points with associated feature set $\mathcal{A}$, where the values of $n_{\mathcal{A}}$ are independent of $j$. Note that $\sum_{c \in [C], \mathcal{A} \subset \mathcal{S}_c} n_{\mathcal{A}} = n$. When training in the $j$-th experiment, we use the cumulative dataset $\mathcal{T}_{1:j} := \bigcup_{l \in [j]} \mathcal{T}_l$, the union of all training datasets up to the $j$-th experiment.

*Remark* 3.2. In each experiment, the feature combinations remain the same across the dataset, but the individual data points differ. This is because each data point is associated with its own specific noise, which varies across samples. Although the underlying features are the same, the noise component of each data point is unique. This approach ensures that the model is exposed to a diverse set of samples.

We define a technical term $h(v; \mathcal{L}) \triangleq \frac{1}{n} \sum_{c \in [C], \mathcal{A} \subset \mathcal{S}_c} n_{\mathcal{A}} \cdot \mathbb{1}(v \in \mathcal{A} \wedge |\mathcal{A} \cap \mathcal{L}| < \tau)$ which denotes the portion of data points containing $v$ that cannot be well-classified by a learned feature set $\mathcal{L}$. This leads to technical assumptions:

**Assumption 3.3.** For any learned feature set $\mathcal{L} \subset \mathcal{S}$, if $v_1, v_2 \in \mathcal{S}_c$ for some class $c \in [C]$ and $h(v_1; \mathcal{L}) = h(v_2; \mathcal{L})$, then $v_1 = v_2$. Also, we assume for any class $c \in [C]$ there exists some $\tau$ distinct features $v_1, \ldots, v_\tau \in \mathcal{S}_c$ such that $g(v_1; \mathcal{T}_j, \mathcal{T}_j), \ldots, g(v_{\tau-1}; \mathcal{T}_j, \mathcal{T}_j) \geq \gamma/n$ and $g(v_\tau; \mathcal{T}_j, \mathcal{T}_j) < \gamma/n$.

This assumption leads to Lemma C.2, stating that the order in which features are learned within a class is deterministic. This is just for simplicity of presentation and can be relaxed. The last assumption is justified by the moderate number of data points in each chunk $\mathcal{T}_j$, ensuring the existence of both $\tau - 1$ learnable features and a non-learnable feature within a class. Throughout the following discussion, we will proceed under above assumptions unless otherwise specified.

### 3.2. Comparison Between Warm-Starting and Cold-Starting in Our Framework

Now we analyze the warm-start and cold-start initialization methods within our framework, focusing on test accuracy and training time. In our learning framework, we denote a model at step $s$ of the $j$-th experiment as $f^{(j,s)}$. We denote the set of learned features and the set of memorized data for the model $f^{(j,s)}$ as $\mathcal{L}^{(j,s)}$ and $\mathcal{M}^{(j,s)}$, respectively. We also denote the set of data points that have non-zero gradients at step $s$ of the $j$-th experiment as $\mathcal{N}^{(j,s)}$. We define two respective versions of these sets and the model, one for warm and cold, denoted by the subscripts (e.g., $f_{\text{warm}}^{(j,s)}$ and $f_{\text{cold}}^{(j,s)}$). We note that, by definition, $\mathcal{L}_{\text{cold}}^{(j,0)}$ and $\mathcal{M}_{\text{cold}}^{(j,0)}$ are both empty sets, while $\mathcal{L}_{\text{warm}}^{(j,0)} = \mathcal{L}_{\text{warm}}^{(j-1,s_{j-1})}$ and $\mathcal{M}_{\text{warm}}^{(j,0)} = \mathcal{M}_{\text{warm}}^{(j-1,s_{j-1})}$, where $s_j$ denotes the last step

of $j$-th experiment. Besides, we use a shorthand notation for step $s_j$ of the experiment $j$ that we drop $s$ if $s = s_j$ (e.g., $\mathcal{L}^{(j)} := \mathcal{L}^{(j,s_j)}$). For the detailed algorithms based on our learning framework, see Algorithms 3 and 4 in Appendix D.

In the test data, a feature combination $\mathcal{A} \subset \mathcal{S}_c$ of data point with class $c \in [C]$ appears with probability $n_{\mathcal{A}}/n$ along with data-specific noise. By Section 2.3, test accuracy for a learned set $\mathcal{L}$ and training time are defined as:

$$\text{ACC}(\mathcal{L}) \triangleq 1 - \frac{C-1}{C} \cdot \frac{1}{n} \sum_{c \in [C], \mathcal{A} \subset \mathcal{S}_c} n_{\mathcal{A}} \cdot \mathbb{1}\left(|\mathcal{A} \cap \mathcal{L}| < \tau\right)$$

$$T_{\text{warm}}^{(J)} \triangleq \sum_{j \in [J]} \left|\mathcal{N}_{\text{warm}}^{(j,0)}\right|, \quad T_{\text{cold}}^{(J)} \triangleq \sum_{j \in [J]} \left|\mathcal{N}_{\text{cold}}^{(j,0)}\right|$$

Based on these definitions, the following theorem holds:

**Theorem 3.4.** *There exists nonempty $\mathcal{G} \subsetneq \mathcal{S}$ such that we always obtain $\mathcal{L}_{\text{warm}}^{(1)} = \mathcal{L}_{\text{cold}}^{(1)} = \mathcal{G}$. For all $J \geq 2$, the following inequalities hold:*

$$\text{ACC}\left(\mathcal{L}_{\text{warm}}^{(J)}\right) \leq \text{ACC}\left(\mathcal{L}_{\text{cold}}^{(J)}\right)$$

$$T_{\text{warm}}^{(J)} < T_{\text{cold}}^{(J)}$$

*Furthermore,* $\text{ACC}(\mathcal{L}_{\text{warm}}^{(J)}) < \text{ACC}(\mathcal{L}_{\text{cold}}^{(J)})$ *holds when* $J > \frac{\gamma}{\delta n}$ *where* $\delta \triangleq \max_{v \in \mathcal{S} \setminus \mathcal{G}} h(v; \mathcal{G}) > 0$.

*Proof Idea.* After the first experiment, the data points in $\mathcal{T}_1$ cannot further contribute to the learning process of the warm-started model. Consequently, even when a new data chunk is provided in subsequent experiments, the feature frequencies are too small, resulting in a weak signal strength of features that cannot overcome the noise signal strength. As a result, the model memorizes individual noise components of the new data points. This procedure is repeated with every experiment, causing the learned feature set to remain *the same* as at the end of the first experiment. In contrast, when receiving $\mathcal{T}_{1:j}$ at once (cold-starting), the signal strength of features is large enough to overcome the noise signal strength, allowing the model to learn many more features. $\square$

Theorem 3.4 highlights a trade-off between cold-starting and warm-starting. Regarding test accuracy, the theorem concludes that cold-starting can achieve strictly higher accuracy than warm-starting. However, warm-starting requires strictly shorter training time compared to cold-starting.

Detailed proof is provided in Appendix C. Theorem 3.4 suggests that the loss of plasticity in the incremental setting under the stationary assumption can be attributed to the noise memorization process. Actually, a similar observation is made in real-world neural network training. It is widely believed that during the early stages of training, neural networks primarily focus on learning features from the dataset,

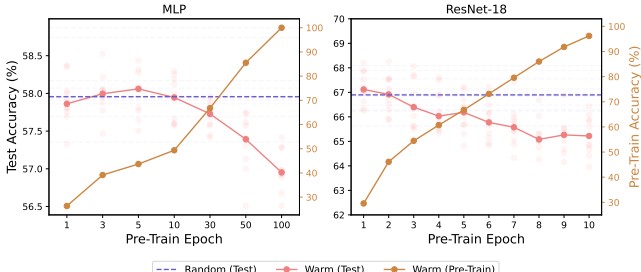

*Figure 2.* The plot shows the test accuracy (left y-axis) when the model is pre-trained for varying epochs (x-axis) and then fine-tuned on the full data, along with the pre-train accuracy (right y-axis) plotted in brown. The left figure shows results from a three-layer MLP, while the right figure presents results from ResNet-18. Each transparent line and point corresponds to a specific random seed, and the median value is clearly displayed. The 'Random' corresponds to training from random initialization (cold-start).

and after learning these features, the model starts to memorize data points that it fails to classify correctly using the learned features. To investigate this phenomenon, we conducted an experiment where CIFAR-10 was divided into two chunks, each containing 50% of the training dataset. The model was pre-trained on one chunk and then further trained on the full dataset for 300 epochs. We used three-layer MLP and ResNet-18 with 10 random seeds.

Figure 2 shows the change in the model's performance based on the duration of pre-training. When pre-training is stopped at a certain epoch and the model is then trained on the full dataset, test accuracy is maintained. However, if pre-training continues beyond a specific threshold (approximately 50% pre-training accuracy in this case), warm-starting significantly impairs the model's performance as it increasingly memorizes training data points. We attribute this phenomenon to the neural network's memorization process after learning features. This is consistent with reports of a critical learning period where neural networks learn useful features in the early phase of learning (Achille et al., 2018; Frankle et al., 2020; Kleinman et al., 2024), and with findings that neural networks tend to learn features followed by memorizing noises (Arpit et al., 2017; Jiang et al., 2020).

*Remark* 3.5. Igl et al. (2020) find that training a model on random labels followed by corrected labels results in better generalization compared to pre-training on a subset of correctly labeled data and then further training on the full dataset with the same distribution. Achille et al. (2018) also observe that pre-training with slightly blurred images followed by original images yields worse test accuracy than pre-training with random label or random noise images. These findings align with our observations: re-training with corrected labels after random label learning "revives" gradi-

ents for most memorized data points, enabling new feature learning. Conversely, with static distributions, gradients for memorized data points remain suppressed, leading to learning from only a few data points with active gradients, causing memorization.

# 4. Proposed Method

## 4.1. Motivation: An Idealized Method

In Section 3, we observed a trade-off between warm-starting and cold-starting. Cold-starting often achieves better test accuracy compared to warm-starting, while warm-starting requires less time to converge. The results suggest that neither retaining all learned information nor discarding all learned information is ideal. To address this trade-off and get the best of both worlds, we consider an idealized algorithm where we retain all learned features while forgetting all memorized data points. For any experiment $J \geq 2$, if we consider the ideal initialization, learned features $\mathcal{L}_{\text{ideal}}^{(J-1)}$ are retained, and memorized data points $\mathcal{M}_{\text{ideal}}^{(J-1)}$ are reset to an empty set. Pseudo-code for this method is given in Algorithm 5, which can be found in Appendix D. We define $T_{\text{ideal}}^{(J)} \triangleq \sum_{j \in [J]} \left| \mathcal{N}_{\text{ideal}}^{(j,0)} \right|$ as the training time with the idealized method, where $\mathcal{N}_{\text{ideal}}^{(j,0)}$ represents the set of data points having a non-zero gradient at the initial step of the $j$-th experiment. Then, we have the following theorem:

**Theorem 4.1.** *For any experiment $J \geq 2$, the following holds:*

$$\text{ACC}\left(\mathcal{L}_{\text{cold}}^{(J)}\right) = \text{ACC}\left(\mathcal{L}_{\text{ideal}}^{(J)}\right)$$
$$T_{\text{warm}}^{(J)} < T_{\text{ideal}}^{(J)} < T_{\text{cold}}^{(J)}$$

The detailed proof is provided in Appendix C. The idealized algorithm addresses the trade-off between cold-starting and warm-starting. We conducted an experiment to investigate the performance gap between these initialization methods.

**Synthetic Experiment.** To see if our theoretical investigation applies more robustly to more realistic scenarios within our framework, we conducted an experiment that more closely resembles real-world settings. Instead of fixing the frequency of each feature set, we sampled each feature's existence from a Bernoulli distribution to construct $\mathcal{V}(\boldsymbol{x})$. This ensures that the experiment is more representative of real-world scenarios. Specifically, for each data point $(\boldsymbol{x}, y)$, we uniformly sampled $y \in \{0, 1\}$. From the feature set $\mathcal{S}_y$ corresponding to the sampled class $y$, we sampled features where each feature's existence follows a Bernoulli distribution, $\mathbb{1}\left(v_{y,k} \in \mathcal{V}(\boldsymbol{x})\right) \sim \text{Ber}(p_k)$, for all $v_{y,k} \in \mathcal{S}_y$. This approach allows us to model the variability in feature occurrence that is commonly observed in real-world datasets while still maintaining the core principles of our learning

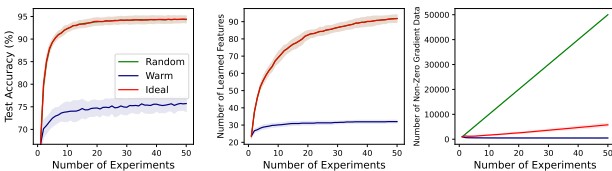

*Figure 3.* Comparison of random, warm, and ideal initialization methods across 10 random seeds (mean ± std dev). The test accuracy (left) and the number of learned features across all classes (middle) are nearly identical for random and ideal initializations, causing their plots to overlap. Warm initialization, however, exhibits lower test accuracy compared to both methods. Regarding training time (right), there is a significant gap between random and warm initialization, which the ideal method addresses.

framework. We set the number of features, $K = 50$, where each $p_k$ is sampled from a uniform distribution, $\text{U}(0, 0.2)$. In each chunk, we sampled 1000 data points and set the total number of experiments to 50, with $\gamma = 50$ and $\tau = 3$. We sampled 10000 test data from the same distribution.

As shown in Figure 3, the results align with the above theorems. Random initialization, i.e. cold-starting, and ideal initialization achieve almost identical generalization performance, outperforming warm initialization. However, with warm initialization, model converges faster, as evidenced by the number of non-zero gradient data points, which serves as a proxy for training time. Ideal initialization requires less time compared to cold-starting, which is also consistent with Theorem 4.1. Due to the sampling process in our experiment, we observe a gradual increase in the number of learned features and test accuracy in warm-starting, mirroring real-world observations. We verified that these results are consistent across a wide range of hyperparameter values (see Figure 14-16 in Appendix B).

## 4.2. DASH: Direction-Aware SHrinking

The aforementioned ideal method recycles memorized training samples by forgetting noise while retaining learned features. This raises the question of whether such an idealized method can be implemented in real-world neural network training. To address this, we propose our algorithm, **Direction-Aware SHrinking (DASH)**, which intuitively captures this idea in practical training scenarios. The outlined behavior is illustrated in Figure 4. In simple terms, DASH shrinks each weight vector proportionally to the cosine similarity between the weight vector and the negative gradient of the loss calculated with train data, with more emphasis on newer data. If the degree of alignment is small (i.e., the cosine similarity is close to or below 0), we consider that the weight vector has not learned a proper feature and shrink it significantly to make it "forget" learned infor-

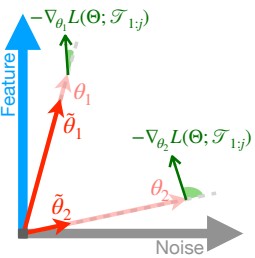

*Figure 4.* Illustration of DASH. We compute the loss $L$ with training data $\mathcal{T}_{1:j}$ and obtain the negative gradient. Then, we shrink the weights proportionally to the cosine similarity between the current weight $\theta$ and $\nabla_\theta L$, resulting in $\tilde{\theta}$.

---

**Algorithm 1** Direction-Aware SHrinking (DASH)

**Require:**
- Model $f_\Theta$ with list of parameters $\Theta$
- Training data points $\mathcal{T}_{1:j}$
- Averaging coefficient $0 < \alpha \leq 1$
- Threshold $\lambda > 0$

**Initialize:**
    $G_\theta^{(0)} \leftarrow 0, \forall \theta$ in $\Theta$

**for** $i$ in $1:j$ **do**
    $\ell \leftarrow \text{Loss}(f_\Theta, \mathcal{T}_i)$
    $U_\Theta \leftarrow$ Gradient of loss $\ell$
    **for** $\theta$ in $\Theta$ **do**
        $G_\theta^{(i)} \leftarrow (1 - \alpha) \times G_\theta^{(i-1)} + \alpha \times U_\theta$
    **end for**
**end for**
**for** $\theta$ in $\Theta$ **do**
    $s_\theta \leftarrow \text{CosineSimilarity}\left(-G_\theta^{(j)}, \theta\right)$
    $\theta \leftarrow \theta \odot \max\{\lambda, s_\theta\}$
**end for**
**return:** $f_\Theta$

---

mation. This allows weights to forget memorized noises and easily change its direction. Conversely, if the weight vector and negative gradient are well-aligned (i.e., the cosine similarity is close to 1), we shrink the weight vector to a lesser degree to maintain the learned information. This intuitive method aligns with the aforementioned idea of the algorithm, as it allows us to shrink weights that have not learned proper information while retaining weights that have learned commonly observed features.

The shrinking is done *per neuron*, where the incoming weights are grouped into a weight vector denoted as $\theta$. For convolutional filters, the height and width of the kernel are flattened to form a single weight vector $\theta$ for each pair of input and output filters. DASH has two hyperparameters: $\lambda$ and $\alpha$. Hyperparameter $\lambda$ is the minimum shrinkage threshold since each weight vector is shrunk by $\max\{\lambda, \cos\_\text{sim}\}$. $\alpha$ is the exponential moving average coefficient for each chunk's gradient. Lower $\alpha$ values emphasize previous gradients, indicating less shrinkage, suitable for less complex datasets where retaining learned features is crucial, and vice versa. The algorithm is presented in Algorithm 1.

To validate whether our intuition aligns well with DASH, we plotted the accuracy on previously learned data points in Figure 12, Appendix B. As experiments progress, when applying DASH, the training accuracy of the previous dataset recovers faster than other methods. We argue that this is due to the nature of our algorithm, which forgets memorized noise while preserving learned features. As the number of experiments increases, the number of learned features also grows. Consequently, we can retain more features compared to previous experiments, resulting in an increase in training accuracy across experiments after applying DASH.

## 5. Experiments

### 5.1. Experimental Details

Our online learning setup is similar to the one described in Ash & Adams (2020). We divided the training dataset into

50 chunks, and at the beginning of each experiment, new data is combined with the existing training data. Models were considered converged and each experiment was terminated when training accuracy reached 99.9%, aligning with our learning framework. Results were averaged over three random seeds. We conducted experiments with vanilla training, i.e., without any data augmentations, weight decay, learning rate scheduling, etc. Additionally, we performed experiments with state-of-the-art (SoTA) settings that include aforementioned techniques. We evaluated DASH on Tiny-ImageNet, CIFAR-10, CIFAR-100, and SVHN using ResNet-18 (He et al., 2016), VGG-16 (Simonyan & Zisserman, 2014), and three-layer MLP architectures with batch normalization layer. Models were trained using Stochastic Gradient Descent (SGD) and SGD based Sharpness-Aware Minimization (SAM), both with momentum.

DASH was compared against baselines (cold-starting, warm-starting, and S&P (Ash & Adams, 2020)) and methods addressing plasticity loss under non-stationarity (L2 INIT (Kumar et al., 2023) and Reset (Nakkiran et al., 2021)). Layer normalization (Ba et al., 2016) and SAM (Foret et al., 2020), known to mitigate plasticity loss in reinforcement learning (Lee et al., 2023), were applied to both warm and cold-starting. Consistent hyperparameters were used across all methods, with details provided in Appendix A.4. S&P, Reset, and DASH were applied whenever new data was introduced. We report two metrics for both test accuracy and steps required for convergence: the value from the final experiment and the average across all experiments.

*Table 1.* Results of training with various datasets using ResNet-18. Bold values indicate the best performance. For the number of steps, bold formatting is used for all methods *except* warm-starting. Results are averaged over three random seeds, with standard deviations provided in parentheses.

| ResNet-18 | Test Acc at Last Experiment | | Number of Steps at Last Experiment | | AVG of Test Acc across All Experiments | | AVG of Number of Steps across All Experiments | |
|---|---|---|---|---|---|---|---|---|
| *T-ImageNet* | SGD | SAM | SGD | SAM | SGD | SAM | SGD | SAM |
| Random Init | 25.69 (0.13) | 31.30 (0.09) | 30237 (368) | 40142 (368) | 17.37 (0.06) | 21.95 (0.11) | 17503 (53) | 22513 (74) |
| Warm Init | 9.57 (0.24) | 13.94 (0.37) | 3388 (368) | 5474 (0) | 6.70 (0.04) | 9.88 (0.21) | 1785 (5) | 2773 (7) |
| S&P | 34.34 (0.48) | 37.39 (0.18) | 13815 (368) | 26066 (1606) | 25.43 (0.02) | 28.47 (0.08) | 7940 (15) | 13172 (182) |
| DASH | **46.11 (0.34)** | **49.57 (0.36)** | **8341 (368)** | **12251 (368)** | **33.06 (0.15)** | **35.93 (0.17)** | **4439 (48)** | **7900 (136)** |
| *CIFAR-10* | | | | | | | | |
| Random Init | 66.75 (0.55) | 75.55 (0.18) | **5213 (184)** | 17734 (184) | 57.82 (0.04) | 66.19 (0.01) | 2889 (24) | **8100 (7)** |
| Warm Init | 64.10 (0.12) | 70.56 (0.30) | 1173 (0) | 4040 (184) | 55.11 (0.10) | 62.94 (0.47) | 726 (29) | 2160 (11) |
| S&P | 81.27 (0.03) | 85.98 (0.18) | 5865 (319) | 32453 (1392) | 71.66 (0.06) | 75.95 (0.13) | **2777 (45)** | 15178 (1462) |
| DASH | **84.28 (0.38)** | **86.88 (0.15)** | 6516 (487) | **13946 (1475)** | **75.01 (0.10)** | **77.93 (0.27)** | 3450 (54) | 10691 (217) |
| *CIFAR-100* | | | | | | | | |
| Random Init | 35.52 (0.10) | 40.16 (0.33) | 10426 (184) | 14336 (184) | 25.65 (0.08) | 29.87 (0.05) | 5762 (76) | 7619 (36) |
| Warm Init | 25.38 (0.64) | 32.02 (0.31) | 1173 (0) | 2346 (0) | 19.34 (0.62) | 24.01 (0.33) | 866 (22) | 1294 (12) |
| S&P | 49.90 (0.09) | 52.94 (0.20) | 4952 (184) | 12251 (1574) | 37.26 (0.15) | 40.27 (0.06) | 2916 (29) | **5896 (162)** |
| DASH | **57.79 (0.22)** | **60.67 (0.20)** | **3519 (0)** | **11339 (0)** | **43.88 (0.05)** | **45.69 (0.12)** | **2027 (58)** | 7184 (641) |
| *SVHN* | | | | | | | | |
| Random Init | 86.49 (0.45) | 89.89 (0.29) | 5474 (0) | **10948 (0)** | 77.99 (0.07) | 83.33 (0.17) | 3099 (14) | **5545 (56)** |
| Warm Init | 84.11 (0.22) | 89.03 (0.22) | 1042 (184) | 1303 (184) | 75.67 (0.27) | 81.30 (0.52) | 630 (6) | 993 (11) |
| S&P | 92.58 (0.17) | 94.29 (0.06) | **3519 (0)** | 11599 (184) | 87.42 (0.13) | 89.37 (0.06) | **1861 (15)** | 5573 (109) |
| DASH | **93.62 (0.03)** | **95.27 (0.02)** | 5083 (844) | 14467 (1105) | **89.63 (0.07)** | **91.69 (0.01)** | 2591 (75) | 9180 (130) |

## 5.2. Experimental Results

We first experimented with CIFAR-10 on ResNet-18 to determine if methods from previous works for mitigating plasticity based on non-stationarity can be a solution to our incremental setting with stationarity. Appendix A.1 shows that L2 INIT, Reset, layer normalization, SAM, and reviving dead neurons, are not effective in our setting. Thus, we conducted the remaining experiments without these methods. Table 1 shows that DASH surpasses cold-starting (Random Init) and S&P in most cases. Training times were often shorter compared to training from scratch, and when longer, the performance gap in test accuracy was more pronounced. Omitted results are in Tables 3-6 located in Appendix A.2.

We argue that S&P can cause the model to forget learned information, including important features, due to shrinking every weight uniformly and perturbing weights. This leads to increased training time and relatively lower test accuracy, especially in SoTA settings (see Appendix A.3). In contrast, DASH preserves properly learned features through direction-aware weight vector shrinkage, addressing these issues.

Theorem 4.1 concludes that ideal initialization can achieve the same test accuracy as cold-starting. However, in real world, DASH surpasses cold-starting in test accuracy. This could be due to the difference between the discrete learning process in our framework and the continuous learning process in real-world neural network training. Even if features have already been learned, DASH can learn them in greater strength compared to learning from scratch, because it preserves the learned features during training. However, in the SoTA setting, different observations are made, which align more closely with our theoretical analysis. These discussions are presented in Appendix A.3.

## 6. Discussion and Conclusion

In this work, we defined an abstract framework for feature learning and discovered that warm-starting benefits from reduced training time compared to random initialization but can hurt the generalization performance of neural networks due to the memorization of noise. Motivated by these observations, we proposed Direction-Aware SHrinking (DASH), which shrinks weights that learned data-specific noise while retaining weights that learned commonly appearing features. We validated DASH in real-world model training, achieving promising results for both test accuracy and training time.

Loss of plasticity is problematic in situations where new data is continuously added daily, which is the case in many real-world application scenarios. Our research aimed to interpret and resolve this issue, preventing substantial wastes in energy, time, and the environment. By elucidating the loss of plasticity phenomenon in stationary data distributions, we have taken a crucial step towards addressing challenges that may emerge in real-world AI, where the continuous influx of additional data is inevitable. We believe our findings contribute to the development of more efficient and sustainable AI that can adapt to ever-increasing real-world data.

We hope our fundamental analysis of the loss of plasticity phenomenon sheds light on understanding this issue as well as providing a remedy. To generalize our findings to any neural network architecture, we treated the learning process as a discrete abstract procedure and did not assume any hypothesis class. Future research could focus on understanding the loss of plasticity phenomenon via optimization or theoretically analyzing it in non-stationary data distributions, such as in reinforcement learning.

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

# A. Experiments

This section provides omitted experiments and hyperparameters used in these experiments. Section A.1 presents the results using non-stationary solutions. The following section compares the experiment results with other baselines, except for those discussed in Appendix A.1. Additionally, we include results conducted on the state-of-the-art (SoTA) setting with CIFAR-10, CIFAR-100 on ResNet-18, in Appendix A.3. In the Appendix A.4, we include hyperparmeters used in these experiments.

The same experimental settings as in Section 5.1 are used. We conducted an experiment with an incremental training dataset comprised of 50 chunks. At the start of each experiment, a new chunk is provided and added to the existing training dataset. Before proceeding to the next experiment, the model is trained until achieving 99.9% train accuracy. Including several baselines, we compared DASH with warm-starting while resetting the momentum at the start of every experiment. We found that this approach cannot be a solution either, as shown in Tables 3-6.

All experiments are conducted using NVIDA-A6000 (48GB VRAM) GPU with two AMD EPYC 7763 64-Core Processor. Number of steps required to converge are provided in each table.

## A.1. Solutions in Non-Stationary Data Distribution

In this subsection, we describe solutions that aim to mitigate plasticity loss under non-stationarity, which cannot remedy the loss of plasticity in an incremental setting with a stationary data distribution. Table 2 shows L2 INIT (Kumar et al., 2023) and Reset (Nikishin et al., 2022) cannot be a solution in our setting.

*Table 2.* Results of training CIFAR-10 dataset trained on various models with solutions proposed to mitigating loss of plasticity in non-stationary data distributions. Bold values indicate the best performance. For the number of steps, we did not provide bold formatting. Results are averaged over three random seeds, with standard deviations provided in parentheses.

| CIFAR-10 | Test Acc at last experiment | | Number of Steps at last experiment | | AVG of Test Acc across all experiments | | AVG of Number of Steps across all experiments | |
|---|---|---|---|---|---|---|---|---|
| *ResNet-18* | SGD | SAM | SGD | SAM | SGD | SAM | SGD | SAM |
| Random Init | **66.75 (0.55)** | **75.55 (0.18)** | 5213 (184) | 17734 (184) | **57.82 (0.04)** | **66.19 (0.01)** | 2889 (24) | 8100 (7) |
| Warm Init | 64.10 (0.12) | 70.56 (0.30) | 1173 (0) | 4040 (184) | 55.11 (0.10) | 62.94 (0.47) | 726 (29) | 2160 (11) |
| L2 INIT | 64.24 (0.80) | 70.32 (0.09) | 1173 (0) | 4040 (184) | 55.47 (0.43) | 62.55 (0.19) | 648 (14) | 2139 (15) |
| Reset | 63.97 (0.45) | 72.03 (0.33) | 1173 (0) | 17986 (1596) | 55.55 (0.30) | 63.40 (0.26) | 976 (51) | 7225 (10) |
| | | | | | | | | |
| *VGG-16* | | | | | | | | |
| Random Init | **84.19 (0.35)** | **86.64 (0.12)** | 21375 (1475) | 37032 (1243) | **75.62 (0.08)** | **77.01 (0.22)** | 12743 (280) | 12509 (343) |
| Warm Init | 78.93 (0.44) | 82.04 (0.04) | 1825 (184) | 4692 (319) | 70.62 (0.24) | 74.00 (0.33) | 1954 (42) | 4277 (315) |
| L2 INIT | 82.79 (0.04) | 82.11 (0.19) | 193936 (58167) | 6126 (665) | 72.11 (0.14) | 73.77 (0.37) | 12489 (443) | 4390 (94) |
| Reset | 78.71 (0.26) | 81.88 (0.35) | 1564 (0) | 3910 (552) | 70.45 (0.36) | 73.31 (0.25) | 1814 (30) | 3230 (51) |
| | | | | | | | | |
| *MLP* | | | | | | | | |
| Random Init | **57.54 (0.31)** | **58.62 (0.13)** | 13555 (184) | 19289 (184) | **51.23 (0.42)** | 52.02 (0.24) | 7516 (166) | 9794 (127) |
| Warm Init | 56.44 (0.33) | 57.67 (0.45) | 2346 (0) | 2216 (184) | 50.60 (0.41) | 51.98 (0.14) | 2309 (408) | 1701 (34) |
| L2 INIT | 56.38 (0.39) | 58.24 (0.06) | 1955 (0) | 2085 (184) | 50.56 (0.51) | **52.15 (0.33)** | 2221 (423) | 1604 (73) |
| Reset | 53.82 (0.32) | 56.42 (0.09) | 6125 (487) | 3389 (184) | 48.89 (0.24) | 50.70 (0.25) | 5955 (740) | 2465 (64) |

Furthermore, applying layer normalization cannot close the gap between cold-starting and warm-starting; rather, the gap increases, as shown in Figure 5. Also, Nikishin et al. (2022) and Sokar et al. (2023) state that loss of plasticity in non-stationary data distributions arises from inactive neurons in the model. However, this is not the case in our setting, as demonstrated in Figure 6.

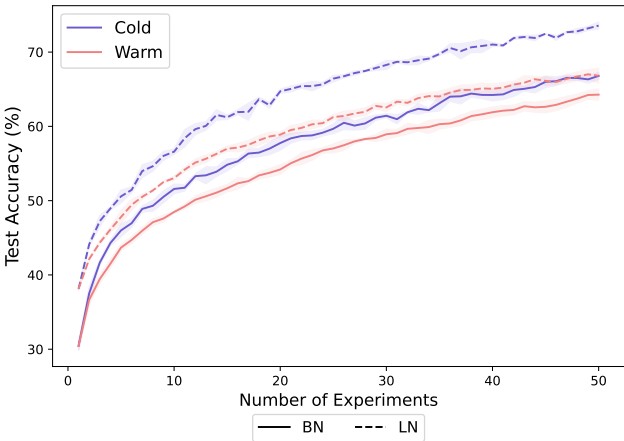

*Figure 5.* The figure shows the results of training ResNet-18 on CIFAR-10 with three random seeds. Layer normalization (dashed lines) is applied in place of batch normalization in ResNet-18, while solid lines represent the use of standard batch normalization. The red lines denote warm-starting, and the blue lines denote cold-starting. The figure demonstrates that the layer normalization technique cannot serve as a solution for plasticity loss. Moreover, the gap between warm-starting and cold-starting performance increases when layer normalization is employed.

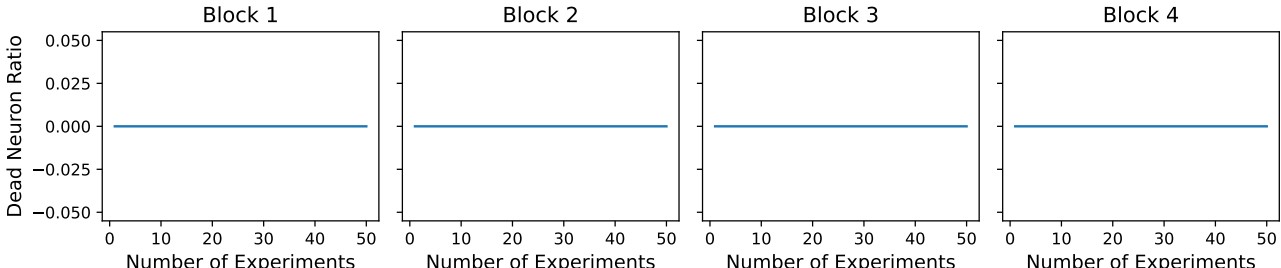

*Figure 6.* The figure presents the results of training ResNet-18 on CIFAR-10 with three random seeds. The presence of dead neurons is assessed after each block of ResNet-18 with training dataset, and the analysis reveals that there are no dead neurons in this case. This finding suggests that techniques designed to revive dead neurons in non-stationary data distributions cannot effectively address the plasticity loss observed in the incremental learning setting with stationary data, which is the primary focus of our study.

## A.2. Omitted Experiment Results

In this subsection, we provide the previously omitted experimental results in Tables 3-6, where the experimental settings are described in Section 5.1. As mentioned earlier, we also compared DASH with warm-starting while resetting the momentum at the start of every experiment, denoted as "Warm ReM" in the following tables.

*Table 3.* Results of training Tiny-ImageNet dataset trained on various models. Bold values indicate the best performance, while underlined values denote the second-best performance in each. For the number of steps, bold formatting is used for all methods except warm-starting. Results are averaged over three random seeds, with standard deviations provided in parentheses.

| T-ImageNet | Test Acc at last experiment | | Number of Steps at last experiment | | AVG of Test Acc across all experiments | | AVG of Number of Steps across all experiments | |
|---|---|---|---|---|---|---|---|---|
| *ResNet-18* | SGD | SAM | SGD | SAM | SGD | SAM | SGD | SAM |
| Random Init | 25.69 (0.13) | 31.30 (0.09) | 30237 (368) | 40142 (368) | 17.37 (0.06) | 21.95 (0.11) | 17503 (53) | 22513 (74) |
| Warm Init | 9.57 (0.24) | 13.94 (0.37) | 3388 (368) | 5474 (0) | 6.70 (0.04) | 9.88 (0.21) | 1785 (5) | 2773 (7) |
| Warm ReM | 9.20 (0.16) | 13.71 (0.29) | 3388 (368) | 5474 (0) | 6.67 (0.08) | 9.93 (0.30) | 1787 (17) | 2795 (14) |
| S&P | 34.34 (0.48) | 37.39 (0.18) | 13815 (368) | 26066 (1606) | 25.43 (0.02) | 28.47 (0.08) | 7940 (15) | 13172 (182) |
| DASH | **46.11 (0.34)** | **49.57 (0.36)** | **8341 (368)** | **12251 (368)** | **33.06 (0.15)** | **35.93 (0.17)** | **4439 (48)** | **7900 (136)** |
| | | | | | | | | |
| *VGG-16* | | | | | | | | |
| Random Init | 40.26 (0.30) | 42.41 (0.13) | 92927 (8940) | 29976 (664) | 28.19 (0.03) | **30.40 (0.04)** | 48878 (799) | 17094 (192) |
| Warm Init | 17.11 (0.44) | 20.77 (0.32) | 1955 (0) | 2997 (184) | 12.91 (0.18) | 15.14 (0.35) | 4359 (162) | 2513 (13) |
| Warm ReM | 17.51 (0.38) | 20.23 (0.06) | 2085 (184) | 2867 (184) | 12.97 (0.24) | 14.87 (0.14) | 4130 (99) | 2472 (8) |
| S&P | 36.56 (0.96) | 38.63 (0.73) | **59432 (5538)** | **18898 (368)** | 23.91 (0.09) | 25.98 (0.22) | **28747 (366)** | **10494 (45)** |
| DASH | **44.29 (0.55)** | **44.40 (0.19)** | 69989 (6215) | 22938 (1329) | **28.47 (0.49)** | 29.11 (0.73) | 31864 (362) | 14258 (149) |
| | | | | | | | | |
| *MLP* | | | | | | | | |
| Random Init | 9.12 (0.06) | 9.19 (0.25) | **28934 (0)** | **42749 (975)** | 6.94 (0.01) | 7.22 (0.03) | **13596 (35)** | **17871 (71)** |
| Warm Init | 7.44 (0.18) | 7.74 (0.25) | 4692 (0) | 4952 (368) | 6.18 (0.03) | 6.41 (0.11) | 2437 (17) | 2797 (38) |
| Warm ReM | 7.54 (0.19) | 7.86 (0.08) | 4431 (368) | 5474 (0) | 6.34 (0.04) | 6.23 (0.05) | 2411 (35) | 2821 (36) |
| S&P | 9.61 (0.22) | 10.28 (0.25) | 33365 (2879) | 55782 (975) | 7.27 (0.01) | 7.57 (0.04) | 16227 (1458) | 21126 (94) |
| DASH | **10.17 (0.19)** | **10.77 (0.12)** | 30237 (975) | 47702 (638) | **7.67 (0.02)** | **8.12 (0.03)** | 17743 (899) | 19455 (212) |

*Table 4.* Results of training CIFAR-10 dataset trained on various models. Bold values indicate the best performance, while underlined values denote the second-best performance. For the number of steps, bold formatting is used for all methods except warm-starting. Results are averaged over three random seeds, with standard deviations provided in parentheses.

| CIFAR-10 | Test Acc at last experiment | | Number of Steps at last experiment | | AVG of Test Acc across all experiments | | AVG of Number of Steps across all experiments | |
|---|---|---|---|---|---|---|---|---|
| *ResNet-18* | SGD | SAM | SGD | SAM | SGD | SAM | SGD | SAM |
| Random Init | 66.75 (0.55) | 75.55 (0.18) | **5213 (184)** | 17734 (184) | 57.82 (0.04) | 66.19 (0.01) | 2889 (24) | **8100 (7)** |
| Warm Init | 64.10 (0.12) | 70.56 (0.30) | 1173 (0) | 4040 (184) | 55.11 (0.10) | 62.94 (0.47) | 726 (29) | 2160 (11) |
| Warm ReM | 64.46 (0.28) | 70.92 (0.39) | 1173 (0) | 3910 (319) | 55.29 (0.43) | 62.85 (0.55) | 727 (51) | 2159 (1) |
| S&P | 81.27 (0.03) | 85.98 (0.18) | 5865 (319) | 32453 (1392) | 71.66 (0.06) | 75.95 (0.13) | **2777 (45)** | 15178 (1462) |
| DASH | **84.28 (0.38)** | **86.88 (0.15)** | 6516 (487) | **13946 (1475)** | **75.01 (0.10)** | **77.93 (0.27)** | 3450 (54) | 10691 (217) |
| | | | | | | | | |
| *VGG-16* | | | | | | | | |
| Random Init | 84.19 (0.35) | 86.64 (0.12) | 21375 (1475) | 37032 (1243) | 75.62 (0.08) | 77.01 (0.22) | 12743 (280) | 12509 (343) |
| Warm Init | 78.93 (0.44) | 82.04 (0.04) | 1825 (184) | 4692 (319) | 70.62 (0.24) | 74.00 (0.33) | 1954 (42) | 4277 (315) |
| Warm ReM | 78.85 (0.40) | 81.41 (0.39) | 1694 (184) | 4692 (319) | 71.02 (0.15) | 73.05 (0.43) | 2097 (123) | 4071 (68) |
| S&P | 85.13 (0.50) | 88.10 (0.19) | 21114 (319) | **36624 (8545)** | 76.52 (0.11) | 79.12 (0.24) | 11794 (213) | **14610 (938)** |
| DASH | **87.58 (0.40)** | **90.52 (0.15)** | **18057 (803)** | 47441 (15391) | **79.92 (0.33)** | **82.93 (0.08)** | **11343 (123)** | 15618 (1049) |
| | | | | | | | | |
| *MLP* | | | | | | | | |
| Random Init | **57.54 (0.31)** | 58.62 (0.13) | 13555 (184) | 19289 (184) | 51.23 (0.42) | 52.02 (0.24) | 7516 (166) | 9794 (127) |
| Warm Init | 56.44 (0.33) | 57.67 (0.45) | 2346 (0) | 2216 (184) | 50.60 (0.41) | 51.98 (0.14) | 2309 (408) | 1701 (34) |
| Warm ReM | 56.26 (0.17) | 57.47 (0.32) | 2215 (184) | 2085 (184) | 50.38 (0.33) | 51.82 (0.33) | 2497 (118) | 1616 (45) |
| S&P | 57.01 (0.43) | 58.03 (0.62) | 6647 (553) | 7038 (319) | 50.98 (0.27) | 52.01 (0.19) | 8228 (1145) | 4262 (83) |
| DASH | 57.20 (0.49) | **58.78 (0.18)** | **6126 (803)** | **6126 (488)** | **51.36 (0.21)** | **52.50 (0.39)** | 7331 (1497) | **3822 (4)** |

*Table 5.* Results of training CIFAR-100 dataset trained on various models. Bold values indicate the best performance, while underlined values denote the second-best performance. For the number of steps, bold formatting is used for all methods except warm-starting. Results are averaged over three random seeds, with standard deviations provided in parentheses.

| CIFAR-100 | Test Acc at last experiment | | Number of Steps at last experiment | | AVG of Test Acc across all experiments | | AVG of Number of Steps across all experiments | |
|---|---|---|---|---|---|---|---|---|
| *ResNet-18* | SGD | SAM | SGD | SAM | SGD | SAM | SGD | SAM |
| Random Init | 35.52 (0.10) | 40.16 (0.33) | 10426 (184) | 14336 (184) | 25.65 (0.08) | 29.87 (0.05) | 5762 (76) | 7619 (36) |
| Warm Init | 25.38 (0.64) | 32.02 (0.31) | 1173 (0) | 2346 (0) | 19.34 (0.62) | 24.01 (0.33) | 866 (22) | 1294 (12) |
| Warm ReM | 24.89 (0.78) | 31.88 (0.73) | 1173 (0) | 2346 (0) | 19.07 (0.72) | 24.03 (0.27) | 817 (33) | 1291 (21) |
| S&P | 49.90 (0.09) | 52.94 (0.20) | 4952 (184) | 12251 (1574) | 37.26 (0.15) | 40.27 (0.06) | 2916 (29) | **5896 (162)** |
| DASH | **57.79 (0.22)** | **60.67 (0.20)** | **3519 (0)** | **11339 (0)** | **43.88 (0.05)** | **45.69 (0.12)** | **2027 (58)** | 7184 (641) |
| | | | | | | | | |
| *VGG-16* | | | | | | | | |
| Random Init | 54.17 (0.51) | 59.14 (0.39) | 63603 (3198) | 26979 (849) | 39.79 (0.11) | 43.78 (0.15) | 29406 (408) | 19153 (217) |
| Warm Init | 36.94 (0.01) | 40.15 (0.61) | 3519 (0) | 4562 (184) | 28.85 (1.10) | 29.82 (0.52) | 4109 (209) | 3646 (139) |
| Warm ReM | 38.59 (0.87) | 39.47 (0.74) | 3388 (184) | 4040 (184) | 29.58 (0.51) | 30.44 (0.51) | 4020 (138) | 3216 (46) |
| S&P | 59.43 (0.16) | **63.50 (0.41)** | 30237 (184) | **11078 (184)** | **45.29 (0.15)** | **47.81 (0.13)** | **14321 (122)** | **7628 (133)** |
| DASH | **59.89 (0.31)** | 62.79 (0.23) | 44704 (567) | 21635 (487) | 43.84 (0.22) | 45.86 (0.36) | 22817 (819) | 12535 (324) |
| | | | | | | | | |
| *MLP* | | | | | | | | |
| Random Init | 28.40 (0.38) | 29.48 (0.40) | 17725 (184) | 25415 (845) | 22.31 (0.08) | 23.56 (0.04) | **11922 (2048)** | **12367 (214)** |
| Warm Init | 26.43 (0.23) | 27.50 (0.14) | 3389 (184) | 2998 (184) | 21.52 (0.11) | 22.44 (0.05) | 5147 (756) | 2727 (98) |
| Warm ReM | 26.18 (0.33) | 27.17 (0.13) | 3910 (552) | 3258 (184) | 21.39 (0.05) | 22.5 (0.08) | 7374 (2251) | 2808 (156) |
| S&P | 30.25 (0.28) | 30.18 (0.13) | **10818 (184)** | 24894 (369) | **23.43 (0.15)** | 23.78 (0.08) | 40492 (5237) | 14400 (502) |
| DASH | **30.31 (0.33)** | **31.21 (0.57)** | 16943 (488) | **21766 (488)** | 23.40 (0.06) | **24.41 (0.07)** | 52789 (4943) | 13379 (467) |

*Table 6.* Results of training SVHN dataset trained on various models. Bold values indicate the best performance, while underlined values denote the second-best performance. For the number of steps, bold formatting is used for all methods except warm-starting. Results are averaged over three random seeds, with standard deviations provided in parentheses.

| **SVHN** | Test Acc at last experiment | | Number of Steps at last experiment | | AVG of Test Acc across all experiments | | AVG of Number of Steps across all experiments | |
|---|---|---|---|---|---|---|---|---|
| *ResNet-18* | SGD | SAM | SGD | SAM | SGD | SAM | SGD | SAM |
| Random Init | 86.49 (0.45) | 89.89 (0.29) | 5474 (0) | **10948 (0)** | 77.99 (0.07) | 83.33 (0.17) | 3099 (14) | **5545 (56)** |
| Warm Init | 84.11 (0.22) | 89.03 (0.22) | 1042 (184) | 1303 (184) | 75.67 (0.27) | 81.30 (0.52) | 630 (6) | 993 (11) |
| Warm ReM | 83.97 (0.12) | 88.89 (0.27) | 782 (0) | 1564 (0) | 75.84 (0.28) | 81.14 (0.59) | 635 (3) | 1010 (7) |
| S&P | 92.58 (0.17) | 94.29 (0.06) | **3519 (0)** | 11599 (184) | 87.42 (0.13) | 89.37 (0.06) | **1861 (15)** | 5573 (109) |
| DASH | **93.62 (0.03)** | **95.27 (0.02)** | 5083 (844) | 14467 (1105) | **89.63 (0.07)** | **91.69 (0.01)** | 2591 (75) | 9180 (130) |
| | | | | | | | | |
| *VGG-16* | | | | | | | | |
| Random Init | 93.65 (0.25) | 93.93 (0.07) | 16552 (1290) | 12251 (184) | 90.45 (0.03) | 90.53 (0.07) | 8496 (204) | 7441 (311) |
| Warm Init | 92.65 (0.23) | 93.20 (0.14) | 1694 (737) | 1042 (184) | 89.60 (0.02) | 89.83 (0.11) | 1116 (42) | 933 (25) |
| Warm ReM | 92.99 (0.16) | 93.28 (0.15) | 1433 (184) | 1042 (184) | 89.61 (0.24) | 89.82 (0.15) | 1116 (22) | 956 (28) |
| S&P | 94.48 (0.21) | **94.87 (0.20)** | **9775 (552)** | **8211 (319)** | 91.84 (0.03) | 91.96 (0.12) | **5956 (129)** | **4898 (13)** |
| DASH | **94.75 (0.19)** | 94.75 (0.17) | 11860 (1208) | 8602 (552) | **91.85 (0.13)** | **92.05 (0.15)** | 6696 (157) | 5502 (8) |
| | | | | | | | | |
| *MLP* | | | | | | | | |
| Random Init | **82.90 (0.04)** | **83.71 (0.32)** | 33756 (664) | 36102 (368) | **77.16 (0.15)** | **78.19 (0.07)** | **19607 (481)** | 18312 (331) |
| Warm Init | 81.19 (0.22) | 82.37 (0.16) | 4692 (319) | 2867 (184) | 76.44 (0.24) | 77.51 (0.08) | 6884 (628) | 2531 (50) |
| Warm ReM | 81.13 (0.39) | 82.25 (0.04) | 4301 (552) | 3128 (319) | 76.49 (0.18) | 77.46 (0.05) | 6470 (693) | 2626 (108) |
| S&P | 82.24 (0.08) | 82.72 (0.34) | 31280 (638) | 16422 (319) | 77.11 (0.05) | 77.93 (0.16) | 20201 (707) | 9786 (278)) |
| DASH | 82.31 (0.39) | 83.18 (0.19) | **25545 (1638)** | **16031 (638)** | 76.81 (0.03) | 77.89 (0.07) | 20062 (178) | **9005 (305)** |

## A.3. Experiments on SoTA setting

In the state-of-the-art (SoTA) setting, we employed weight decay and standard data augmentation techniques, such as horizontal flipping and random cropping. We also used a learning rate scheduler that reduces the learning rate step-wise by a factor of 0.2 at 60, 120, and 200 epochs. By applying the learning rate scheduler, there is no need to compare training time since training is completed at roughly the same epoch across all experiments. The weight decay was set to 0.0005, and the initial learning rate was set to 0.1. All other settings remain the same as mentioned above. We tested this setup on CIFAR-10 and CIFAR-100 using the ResNet-18 architecture.

The results in Table 7 show that DASH performs similarly to or slightly worse than starting from random initialization. It appears that this is partly because all hyperparameters are tuned to maximize the performance of cold-starting, to achieve the (close-to-)SoTA test accuracy numbers. Due to the lack of computational resources, we were unable to tune hyperparameters specifically for DASH.

Furthermore, we believe this aligns more closely with our theoretical anylsis in Theorem 4.1, as the hyperparameters are tuned to allow the model to learn as many features as possible, making it difficult for DASH to outperform cold-starting.

Moreover, we observe that S&P cannot be used in these SoTA settings. We believe this is due to the nature of S&P, which shrinks all weights, while the SoTA setting is likely designed to avoid learning unuseful features, unlike the previous setting. Consequently, it is plausible that retaining learned features is more important than forgetting them, making S&P unsuitable for SoTA settings. Although DASH performs slightly worse than cold-starting, it is conceivable that it is better at retaining features compared to S&P and other warm-starting methods, resulting in better overall performance.

The gap between warm-starting and cold-starting has been significantly reduced, likely due to data augmentation techniques and the increase in learning rate when new data is introduced. Data augmentation techniques increase the amount of feature information, allowing warm-starting to learn features that vanilla training (without augmentation) cannot (Shen et al., 2022). Furthermore, as the learning rate is set to a higher value at the beginning of each new experiment, the model can forget previously memorized data points and escape spurious minima that were difficult to escape from, which is consistent with the findings of Berariu et al. (2021). Despite these improvements, a gap still exists between warm-starting and cold-starting.

*Table 7.* Results of training CIFAR-10, CIFAR-100 dataset trained on ResNet-18 with SoTA settings. Bold values indicate the best performance, while underlined values denote the second-best performance. For the number of steps, we did not provide bold formatting since we used learning rate scheduling. Results are averaged over three random seeds, with standard deviations provided in parentheses.

| *ResNet-18* | Test Acc at last experiment | | Number of Steps at last experiment | | AVG of Test Acc across all experiments | | AVG of Number of Steps across all experiments | |
|---|---|---|---|---|---|---|---|---|
| **CIFAR-10** | SGD | SAM | SGD | SAM | SGD | SAM | SGD | SAM |
| Random Init | **94.73 (0.14)** | **95.47 (0.17)** | 50439 (319) | 47832 (184) | **88.77 (0.04)** | 89.24 (0.15) | 24826 (62) | 23751 (34) |
| Warm Init | 94.35 (0.31) | 94.80 (0.20) | 51221 (552) | 47832 (184) | 87.94 (0.26) | 88.62 (0.57) | 23759 (57) | 21821 (174) |
| Warm ReM | 94.56 (0.25) | 95.00 (0.29) | 51612 (319) | 47962 (184) | 88.20 (0.33) | 88.56 (0.60) | 23775 (16) | 21786 (79) |
| S&P | 94.15 (0.10) | 94.73 (0.07) | 51351 (184) | 48353 (184) | 88.38 (0.03) | 89.27 (0.26) | 25369 (49) | 22805 (16) |
| DASH | 94.25 (0.25) | 95.06 (0.36) | 51872 (487) | 48223 (184) | 88.65 (0.24) | **89.34 (0.40)** | 24264 (75) | 22233 (85) |
| | | | | | | | | |
| **CIFAR-100** | | | | | | | | |
| Random Init | **75.98 (0.01)** | **76.09 (0.12)** | 63081 (184) | 56825 (184) | **61.49 (0.09)** | **61.81 (0.08)** | 27536 (194) | 25521 (91) |
| Warm Init | 74.10 (0.09) | 74.21 (0.26) | 69598 (1462) | 58128 (921) | 58.40 (0.24) | 58.44 (0.12) | 28012 (114) | 24562 (243) |
| Warm ReM | 74.05 (0.13) | 74.36 (0.13) | 68425 (1689) | 57216 (664) | 58.32 (0.24) | 58.33 (0.15) | 27965 (190) | 24534 (139) |
| S&P | 72.96 (0.34) | 73.71 (0.37) | 64775 (664) | 61387 (552) | 57.33 (0.10) | 57.68 (0.06) | 28809 (148) | 26476 (212) |
| DASH | 74.84 (0.07) | 74.98 (0.09) | 67121 (1815) | 59953 (1208) | 60.89 (0.20) | 61.29 (0.13) | 28746 (306) | 25630 (100) |

### A.4. Hyperparameters

In this section, we will provide the hyperparameters utilized in our experiments. Additionally, we present heatmaps illustrating the results for a wide range of two hyperparameters, $\alpha$ and $\lambda$, in DASH. The heatmaps in Figures 7 and 8 suggest that DASH exhibits robustness to hyperparameter variations, indicating that its performance is less affected by the choice of hyperparameter values.

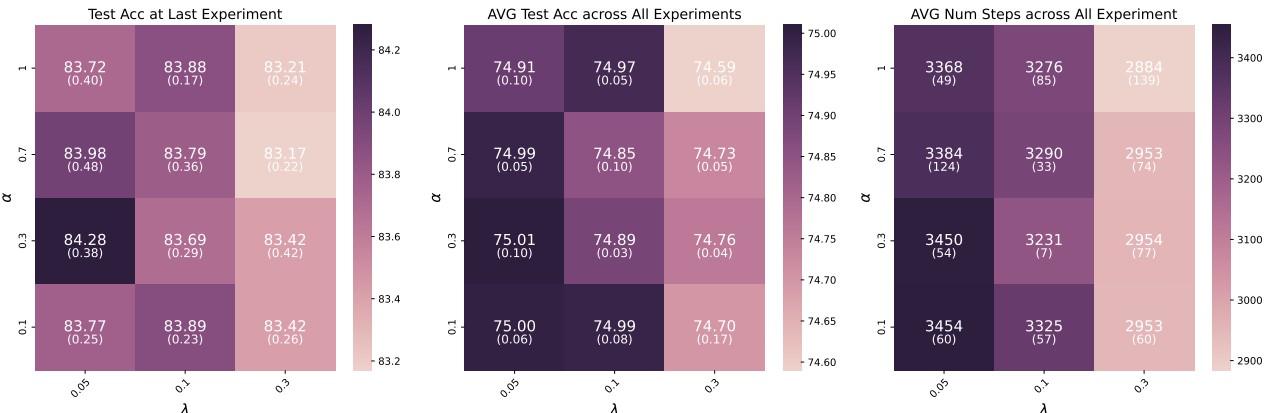

*Figure 7.* Heatmap illustrating the performance of DASH with various hyperparameter values on the CIFAR-10 dataset using a ResNet-18 architecture. Three runs averaged with standard deviation. Darker colors indicate higher values. The first two heatmaps show that higher values are preferable, while the last heatmap demonstrates that lower values (brighter colors) are preferable.

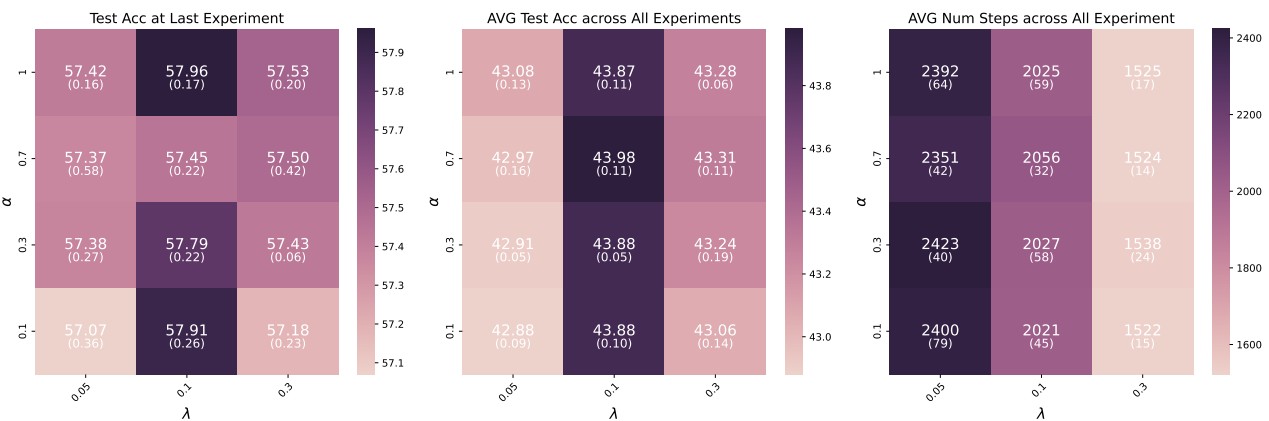

*Figure 8.* Heatmap illustrating the performance of DASH with various hyperparameter values on the CIFAR-100 dataset using a ResNet-18 architecture. Three runs averaged with standard deviation. Darker colors indicate higher values. The first two heatmaps show that higher values are preferable, while the last heatmap demonstrates that lower values (brighter colors) are preferable.

We fixed the momentum to 0.9 and the batch size to 128. The learning rate is set to 0.001 for training ResNet-18, and for other models, a learning rate of 0.01 is used. The value of $\rho$ for SAM is chosen based on the performance of cold-starting. The default value of $\alpha = 0.3$ is used, and we did not change this value frequently. The perturbation parameter $\sigma$ used in the Shrink & Perturb (S&P) procedure is set to $0.01$, as this value is considered optimal for perturbation, as described in Ash & Adams (2020). Initially, we tested $\sigma = 0.1$ as the perturbation parameter, since Ash & Adams (2020) reported slightly better test accuracy compared to $\sigma = 0.01$ in some cases. However, we experienced significantly poorer generalization performance with $\sigma = 0.1$ compared to $\sigma = 0.01$, as shown in Figure 9. The hyperparameters used in our experiments are described in Table 8.

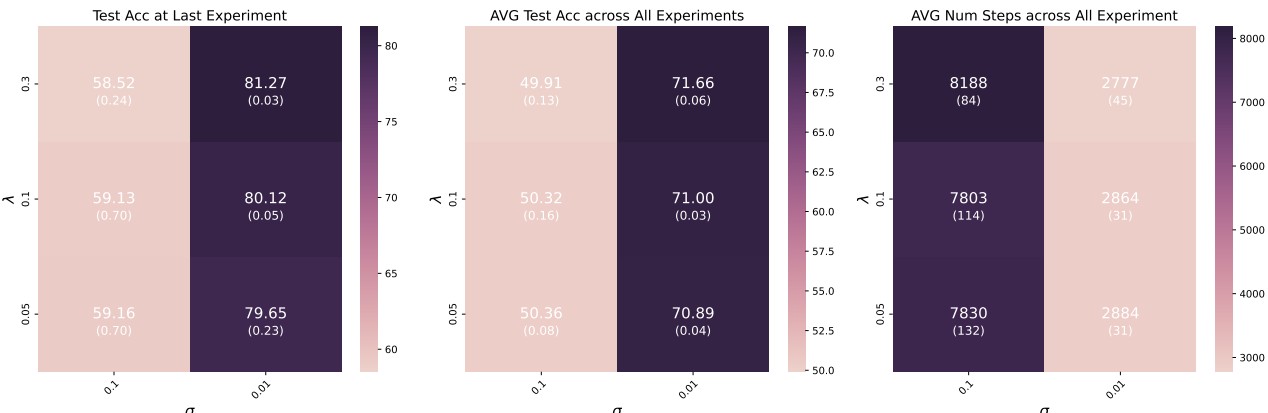

*Figure 9.* Heatmap showing the performance of S&P on CIFAR-10 using ResNet-18 with varying $\sigma$ values. While Ash & Adams (2020) reported better test accuracy when $\sigma = 0.1$ compared to $\sigma = 0.01$, we exhibited significantly lower performance compared to $\sigma = 0.01$.

*Table 8.* The hyperparameters used in our experiments are shown in the table, where the values on the left side of the '/' correspond to those used for the SGD optimizer, and the values on the right side correspond to those used for the SAM optimizer. In the case of Shrink & Perturb (S&P), the $\lambda$ value corresponds to the shrinkage parameter, while the $\sigma$ parameter controls the magnitude of the noise added to the weights after shrinking. For L2 INIT, we did not perform experiments except for CIFAR-10, hence the values are omitted for other datasets.

| | | | | | DASH | | S&P | | L2 INIT |
|---|---|---|---|---|---|---|---|---|---|
| **ResNet-18** | Momentum | LR | Batch Size | $\rho$ | $\lambda$ | $\alpha$ | $\lambda$ | $\sigma$ | $\lambda$ |
| Tiny-Imagenet | 0.9 | 0.001 | 128 | 0.05 | 0.05 | 0.3 | 0.05 | 0.01 | - |
| CIF1R-10 | 0.9 | 0.001 | 128 | 0.1 | 0.05/0.3 | 0.3 | 0.3 | 0.01 | 1e-4 |
| CIF1R-100 | 0.9 | 0.001 | 128 | 0.05 | 0.1 | 0.3 | 0.3 | 0.01 | - |
| SVHN | 0.9 | 0.001 | 128 | 0.05 | 0.3 | 0.3 | 0.3 | 0.01 | - |
| **VGG16** | | | | | | | | | |
| Tiny-Imagenet | 0.9 | 0.01 | 128 | 0.05 | 0.05 | 0.3 | 0.05 | 0.01 | - |
| CIF1R-10 | 0.9 | 0.01 | 128 | 0.1 | 0.05/0.1 | 0.3 | 0.1 | 0.01 | 1e-4 |
| CIF1R-100 | 0.9 | 0.01 | 128 | 0.03 | 0.05 | 0.9/0.3 | 0.3 | 0.01 | - |
| SVHN | 0.9 | 0.01 | 128 | 0.05 | 0.1 | 0.9/0.3 | 0.3 | 0.01 | - |
| **MLP** | | | | | | | | | |
| Tiny-Imagenet | 0.9 | 0.01 | 128 | 0.1 | 0.1 | 0.3 | 0.3/0.1 | 0.01 | - |
| CIF1R-10 | 0.9 | 0.01 | 128 | 0.1 | 0.7/0.5 | 0.3 | 0.7/0.5 | 0.01 | 1e-4 |
| CIF1R-100 | 0.9 | 0.01 | 128 | 0.1 | 0.1 | 0.3 | 0.3/0.1 | 0.01 | - |
| SVHN | 0.9 | 0.01 | 128 | 0.1 | 0.3 | 0.3 | 0.3 | 0.01 | - |

## B. Justification of Feature Learning Process

This section justifies our proposed learning framework proposed in Section 2.

Figure 10 shows that the initial gradient norm of training data, $|\mathcal{N}^{(j,0)}|$, can be a proxy for the training time until convergence. As the initial gradient norm increases, the number of steps required for convergence also increases. While this figure uses the gradient norm instead of the number of non-zero gradient data points due to the continuous nature of real-world neural network training, we believe it resembles the behavior of non-zero gradient data points. Additionally, Figure 11 demonstrates that the number of steps required for convergence increases as the number of data points increases in various datasets.

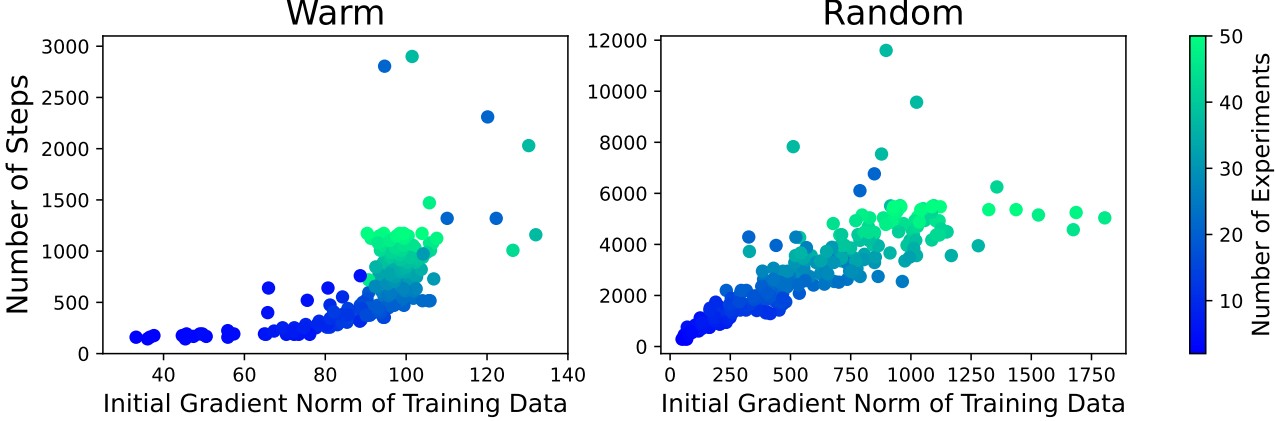

*Figure 10.* Trained on ResNet-18 with five random seeds, where CIFAR-10 is divided into 50 chunks and incrementally increased by adding new chunks at each experiment. Each point represents an individual experiment. The gradient norm is used as a proxy for the number of non-zero gradient data points, which in turn serves as a proxy for the training time. A larger gradient norm indicates the model needs to learn more features or memorize more data points to correctly classify all training data points.

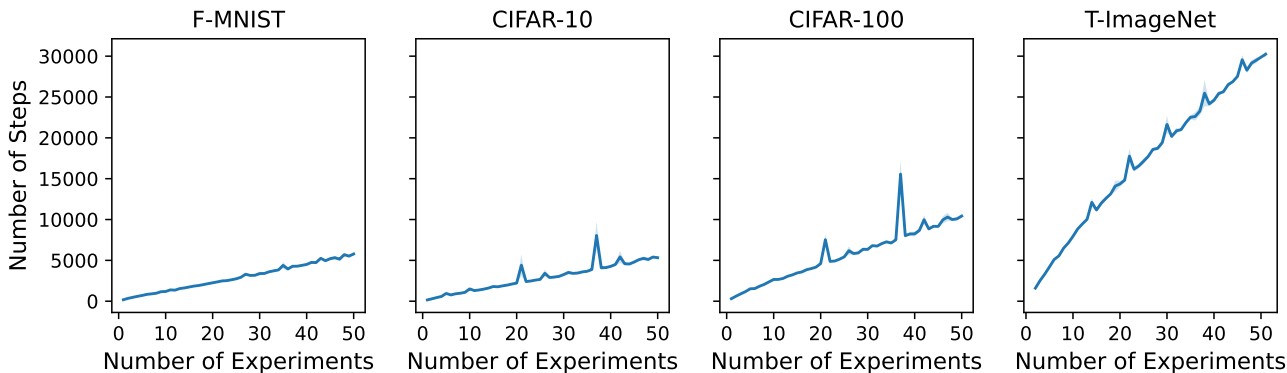

*Figure 11.* Figure trained on ResNet-18 with three random seeds. The dataset is divided into 50 chunks, and new chunks are incrementally added for each experiment. The number of steps required for convergence increases with the amount of data when training a cold-started neural network, which is a standard training.

To validate whether previously learned/memorized data points do not have large gradients when further trained with a combined dataset (existing + newly introduced data) using warm-starting, we plotted the train accuracy on the previous dataset for the first few epochs using CIFAR-10 trained on ResNet-18 (Figure 12). We observe that the newly introduced data does not substantially affect the previously learned/memorized data points, even when further trained with warm-starting, supporting the main idea of Theorem 3.4.

To verify whether DASH truly captures our intuitions from the ideal algorithms, we conducted an experiment using CIFAR-10 trained on ResNet-18, with the same experimental settings. Figure 12 demonstrates that when applying DASH, the train accuracy on previous datasets rapidly increases after a few epochs compared to other methods. We argue that this behavior stems from our algorithm's ability to forget memorized noise while preserving learned features. As the number of experiments increases, the number of learned features also grows. For a fair comparison, we used $\lambda = 0.05$ for DASH, and when performing S&P and shrink, we shrank each weight by multiplying $0.05$. In the case of S&P, after shrinking, we added noise sampled from $\mathcal{N}(0, 0.01^2)$.

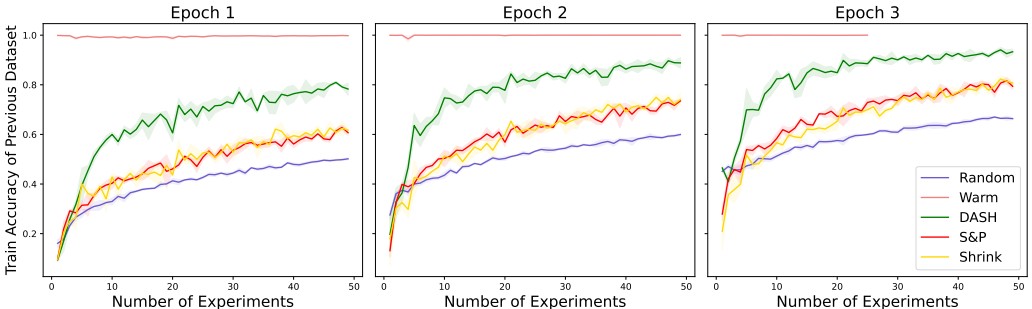

*Figure 12.* The results are averaged over 10 random seeds. The x-axis represents the number of experiments, while the y-axis represents the training accuracy on previous datasets. Warm-starting can retain previously learned data points when further trained with an incremented dataset. Additionally, DASH, plotted in green, can retain more information compared to other methods.

As stated in Section 2, we posited that $\tau$ could serve as a proxy for dataset complexity. Figure 14 shows that as $\tau$ increases, the threshold for considering a data point well-classified also increases, making it more difficult to correctly predict unseen data points. This difficulty is particularly pronounced for warm-starting, leading to a widening gap between the random initialization and warm initialization methods. Additionally, this phenomenon is observed in real-world neural network training, as depicted in Figure 13. For datasets with higher complexity (from left to right), the gap between the two initialization methods widens, exhibiting the same trend as an increasing $\tau$.

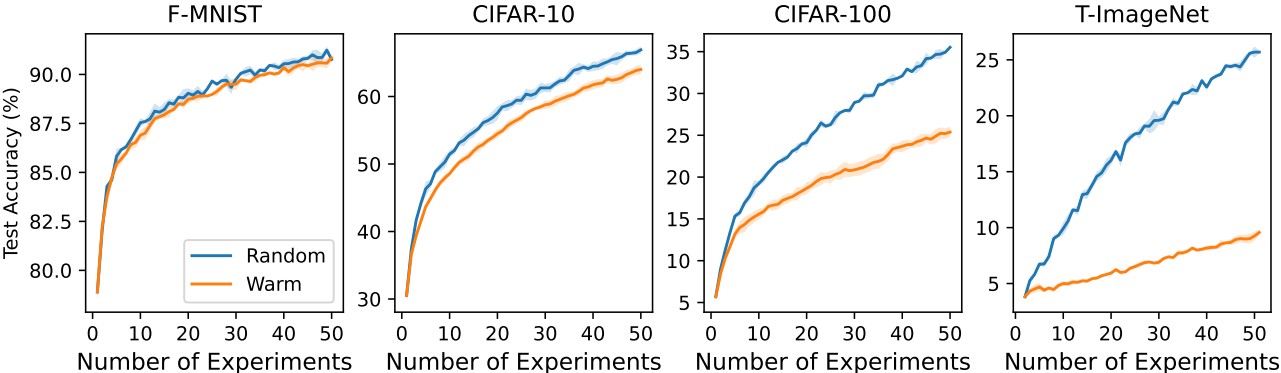

*Figure 13.* The same hyperparameters are used described in Section 5.1 with three random seeds. The gap in test accuracy between the two initialization methods increases as dataset complexity increases.

We conducted synthetic experiments across a wide range of hyperparameters. Figure 14 uses the same setup as Section 3 but varies $\tau$. Figure 15 explores different numbers of classes, $C$, while Figure 16 investigates varying noise signal strengths, $\gamma$. These results align with our findings from Theorems 3.4 and 4.1.

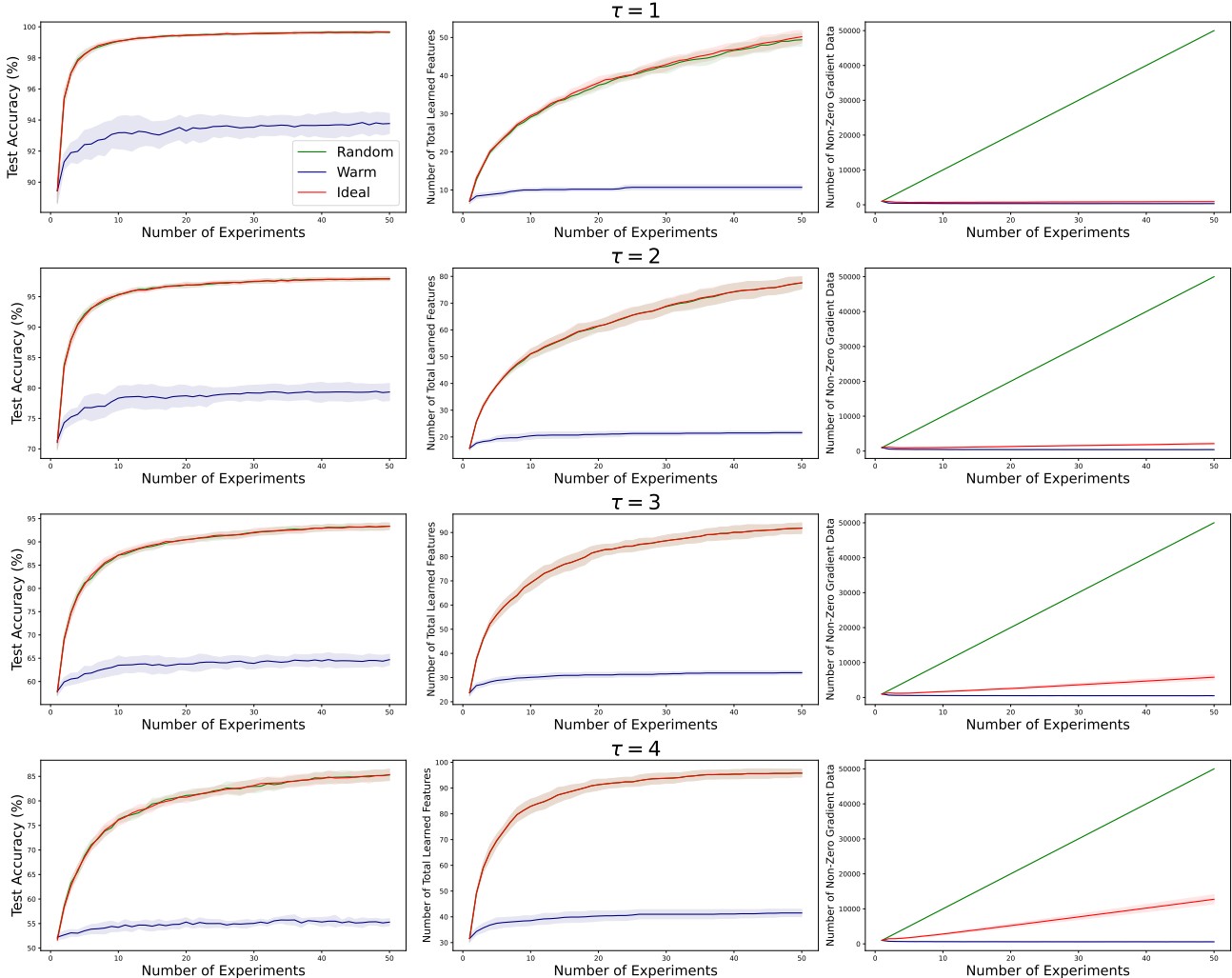

*Figure 14.* Results using the same hyperparameters as described in Section 3, except for the threshold for a data point is considered well-classified ($\tau$). Experiments were conducted with 10 random seeds. The trend observed in Figure 3 persists across different values of $\tau$.

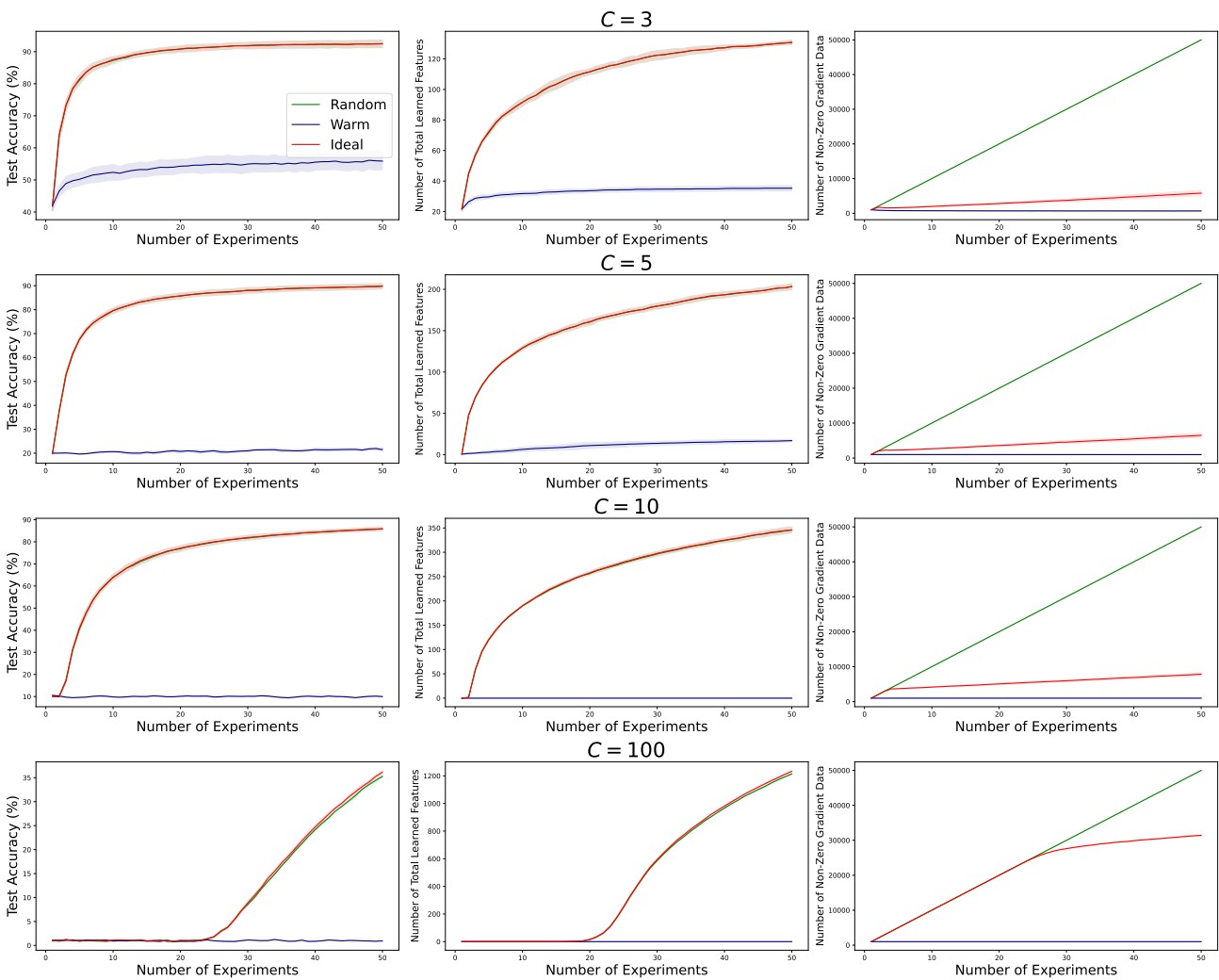

*Figure 15.* Results using the same hyperparameters as described in Section 3, except for the number of classes ($C$). Experiments were conducted with 10 random seeds. The trend observed in Figure 3 persists across different values of $C$.

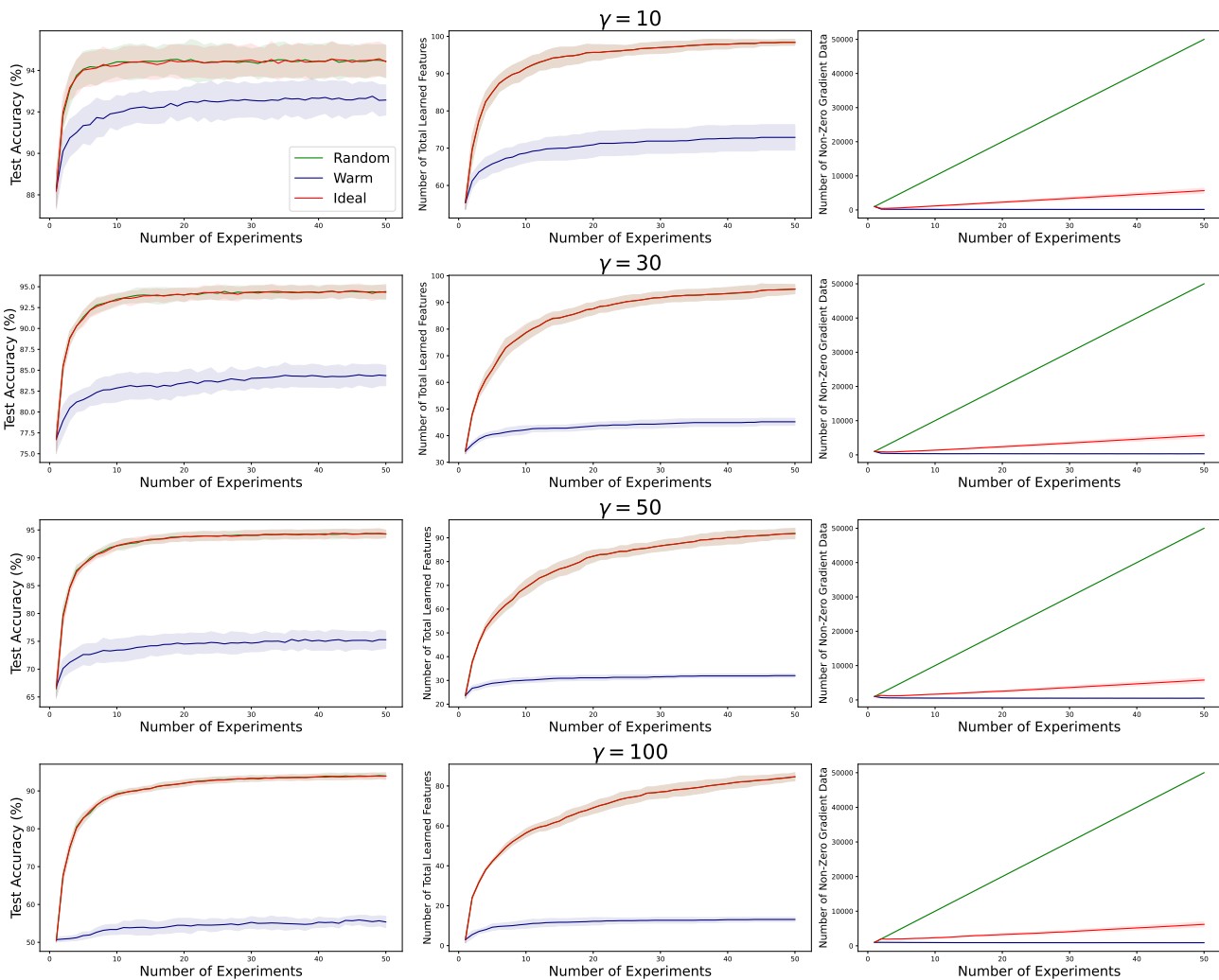

*Figure 16.* Results using the same hyperparameters as described in Section 3, except for the strength of the noise, $\gamma$. Experiments were conducted with 10 random seeds. The trend observed in Figure 3 persists across different values of $\gamma$.

## C. Proof of Theorems

This section provides the proof for Theorems 3.4 and 4.1, stated in Sections 3 and 4, respectively. Before presenting the main proof, we state some technical lemmas.

**Lemma C.1.** *For any learned feature set $\mathcal{A}, \mathcal{B} \subset \mathcal{S}$ that satisfies $\mathcal{A} \subsetneq \mathcal{B}$ and $|\mathcal{A} \cap \mathcal{S}_c| \geq \tau - 1$ for any $c \in [C]$, then we have $\mathrm{ACC}(\mathcal{A}) < \mathrm{ACC}(\mathcal{B})$.*

*Proof of Lemma C.1.* Since $\mathcal{A} \subsetneq \mathcal{B}$, it is trivial that for any $c \in [C]$ and $\Lambda \subset \mathcal{S}_c$, we have

$$\mathbb{1}\left(|\Lambda \cap \mathcal{A}| < \tau\right) \geq \mathbb{1}\left(|\Lambda \cap \mathcal{B}| < \tau\right). \tag{1}$$

From the given condition, we can choose $c^* \in [C]$ such that there exists $\tau - 1$ distinct features $v_1, \ldots, v_{\tau-1} \in \mathcal{A} \cap \mathcal{S}_{c^*}$ and $v_\tau \in (\mathcal{B} \cap \mathcal{S}_{c^*}) \setminus (\mathcal{A} \cap \mathcal{S}_{c^*})$. Our choice of $\Lambda^* \triangleq \{v_1, \ldots, v_\tau\} \subset \mathcal{S}_{c^*}$ satisfies

$$\mathbb{1}\left(|\Lambda^* \cap \mathcal{A}| < \tau\right) > \mathbb{1}\left(|\Lambda^* \cap \mathcal{B}| < \tau\right). \tag{2}$$

From (1), (2), and the definition of $\mathrm{ACC}(\cdot)$, we have

$$\mathrm{ACC}(\mathcal{A}) = 1 - \frac{C-1}{C} \cdot \frac{1}{n} \sum_{c \in [C], \Lambda \subset \mathcal{S}_c} n_\Lambda \cdot \mathbb{1}\left(|\Lambda \cap \mathcal{A}| < \tau\right) < 1 - \frac{C-1}{C} \cdot \frac{1}{n} \sum_{c \in [C], \Lambda \subset \mathcal{S}_c} n_\Lambda \cdot \mathbb{1}\left(|\Lambda \cap \mathcal{B}| < \tau\right) = \mathrm{ACC}(\mathcal{B}).$$

$\square$

For ease of presentation, let us say "a model learns $\mathrm{NA}$" if a model cannot learn any features. For example, if a model learns $u_1, \cdots, u_s \in \mathcal{S}$ during $s$ steps of training process and feature learning process ends in $(s+1)$-th step, let us say that we learn $u_1, \cdots, u_s, \mathrm{NA}, \mathrm{NA}, \cdots$.

Using the notion above, we prove that our learning process uniquely determines the behavior within the same class regardless of the randomness of the training process, where the randomness may come from tie-breaking that can happen in the choice of the most frequent non-learned feature.

**Lemma C.2.** *Suppose we train two models with different randomness on $\mathcal{T}_{1:j}$ for some $j \in \mathbb{N}$ starting from a learned set $\mathcal{L}$ and without any memorized data. We use $u_s$ and $u'_s$ to denote features learned in $s$-th step of training process by two models, respectively. The $i$-th learned feature within class $c \in [C]$ is denoted as $u_{c,i}$ for the first model and $u'_{c,i}$ for the second model. Then, $u_{c,i} = u'_{c,i}$ for all $c \in [C]$ and $i \in \mathbb{N}$.*

*Proof of Lemma C.2.* Suppose there exists some class $c \in [C]$ and $i \in \mathbb{N}$ such that $u_{c,i} \neq u'_{c,i}$ and choose one with the smallest $i$. Without loss of generality, we may assume $u'_{c,i} \neq \mathrm{NA}$. Then, we have

$$\max_{v \in \mathcal{S}_c \setminus \{u_{c,1}, \ldots, u_{c,i-1}\}} h(v; \{u_{c,1}, \ldots, u_{c,i-1}\}) = \max_{v \in \mathcal{S}_c \setminus \{u'_{c,1}, \ldots, u'_{c,i-1}\}} h(v; \{u'_{c,1}, \ldots, u'_{c,i-1}\})$$
$$= h(u'_{c,i}; \{u'_{c,1}, \ldots, u'_{c,i-1}\})$$
$$\geq \frac{\gamma}{jn}$$

The first equality holds since $\{u_{c,1}, \ldots, u_{c,i-1}\} = \{u'_{c,1}, \ldots, u'_{c,i-1}\}$ from our choice of $c, i$ and the second equality holds since the second model learns $u'_{c,i}$. The last inequality holds since $u'_{c,i} \neq \mathrm{NA}$. Hence, $u_{c,i} \neq \mathrm{NA}$ and

$$h(u_{c,i}; \{u_{c,1}, \ldots, u_{c,i-1}\}) = \max_{v \in \mathcal{S}_c \setminus \{u_{c,1}, \ldots, u_{c,i-1}\}} h(v; \{u_{c,1}, \ldots, u_{c,i-1}\}).$$

From our Assumption 3.3 and since $\{u_{c,1}, \ldots, u_{c,i-1}\} = \{u'_{c,1}, \ldots, u'_{c,i-1}\}$, we have $u_{c,i} = u'_{c,i}$. This is contradictory and we have our desired conclusion. $\square$

**Lemma C.3.** *Suppose we train two models on $\mathcal{T}_{1:j_1}$ and $\mathcal{T}_{1:j_2}$ for some $j_1 > j_2$ starting from a learned set $\mathcal{L}$ and without any memorized data. We use $u_s$ and $u'_s$ to denote features learned in $s$-th step of the training process by two models trained on $\mathcal{T}_{1:j_1}$ and $\mathcal{T}_{1:j_2}$, respectively. The $i$-th learned feature within class $c \in [C]$ is denoted as $u_{c,i}$ for the first model and $u'_{c,i}$ for the second model. Then, $u_{c,i} = u'_{c,i}$ or $u'_{c,i} = NA$ for all $c \in [C]$ and $i \in \mathbb{N}$.*

*Proof of Lemma C.3.* Suppose there exists some class $c \in [C]$ and $i \in \mathbb{N}$ such that $u_{c,i} \neq u'_{c,i}$ and $u'_{c,i} \neq \texttt{NA}$. Choose one with the smallest $i$. Since $u'_{c,i} \neq \texttt{NA}$ and from our choice of $c$ and $i$, we have

$$
\max_{v \in \mathcal{S}_c \setminus \{u_{c,1},\ldots,u_{c,i-1}\}} h(v; \{u_{c,1},\ldots,u_{c,i-1}\}) = \max_{v \in \mathcal{S}_c \setminus \{u'_{c,1},\ldots,u'_{c,i-1}\}} h(v; \{u'_{c,1},\ldots,u'_{c,i-1}\})
$$
$$
= h(u'_{c,i}; \{u'_{c,1},\ldots,u'_{c,i-1}\})
$$
$$
\geq \frac{\gamma}{j_2 n} > \frac{\gamma}{j_1 n}.
$$

Hence, $u_{c,i} \neq \texttt{NA}$ and

$$
h(u_{c,i}; \{u_{c,1},\ldots,u_{c,i-1}\}) = \max_{v \in \mathcal{S}_c \setminus \{u_{c,1},\ldots,u_{c,i-1}\}} h(v; \{u_{c,1},\ldots,u_{c,i-1}\}).
$$

From our Assumption 3.3 and since $\{u_{c,1},\ldots,u_{c,i-1}\} = \{u'_{c,1},\ldots,u'_{c,i-1}\}$, we have $u_{c,i} = u'_{c,i}$. This is contradictory and we have our desired conclusion. $\square$

With above Lemma C.1, C.2 and C.3, we have the following theorems.

### C.1. Proof of Theorem 3.4

By Lemma C.2 for the case $j = 1$, we immediately have our first conclusion by defining $\mathcal{G}$ as a learned feature set from the first experiment. Furthermore, we have $\mathcal{G} \subsetneq \mathcal{S}_c$ and $|\mathcal{G} \cap \mathcal{S}_c| \geq \tau - 1$ for any class $c \in [C]$ from our feature learning framework and Assumption 3.3.

We want to show that for any $J \geq 2$, $\mathcal{L}_{\text{warm}}^{(J)} = \mathcal{G}$. Since we never forget the learned feature in warm training, it is clear that $\mathcal{G} = \mathcal{L}_{\text{warm}}^{(1)} \subset \mathcal{L}_{\text{warm}}^{(J)}$. We may assume that the existence of $J^* \geq 2$ such that $\mathcal{L}_{\text{warm}}^{(1)} \subsetneq \mathcal{L}_{\text{warm}}^{(J^*)}$ and choose the smallest $J^* \geq 2$. Then, in the first step of $J^*$-th experiment, a model learns some feature $u$. From our training process, $u$ satisfies

$$
n \cdot h(u; \mathcal{G}) = |\mathcal{T}_{1:J^*}| \cdot g(u; \mathcal{T}_{1:J^*}, \mathcal{N}_{\text{warm}}^{(J^*,0)}) \geq \gamma,
$$

and since $J^*$ denotes the first experiment that can learn beyond $\mathcal{G}$, $\mathcal{L}_{\text{warm}}^{(J^*-1)} = \mathcal{G}$ and

$$
|\mathcal{T}_{1:J^*-1}| \cdot g(u; \mathcal{T}_{1:J^*-1}, \mathcal{N}_{\text{warm}}^{(J^*-1,0)}) = n \cdot h(u; \mathcal{G}) \geq \gamma.
$$

It means that $u$ must have been already learned in the $(J^* - 1)$-th experiment and it is contradictory.

Thus, we have $\mathcal{L}_{\text{warm}}^{(J)} = \mathcal{L}_{\text{warm}}^{(1)} = \mathcal{L}_{\text{cold}}^{(1)} \subset \mathcal{L}_{\text{cold}}^{(J)}$ for all $J \geq 2$ and combining with Lemma C.1, we have

$$
\text{ACC}(\mathcal{L}_{\text{warm}}^{(J)}) = \text{ACC}(\mathcal{L}_{\text{warm}}^{(1)}) = \text{ACC}(\mathcal{L}_{\text{cold}}^{(1)}) \leq \text{ACC}(\mathcal{L}_{\text{cold}}^{(J)}).
$$

To show that strict inequality for $J > \frac{\gamma}{\delta n}$, it suffices to show that $\mathcal{L}_{\text{cold}}^{(1)} \subsetneq \mathcal{L}_{\text{cold}}^{(J)}$ for $J > \frac{\gamma}{\delta n}$ since we already showed that $\mathcal{G} \subsetneq \mathcal{S}$ and $|\mathcal{G} \cap \mathcal{S}_c| \geq \tau - 1$ for any class $c \in [C]$. In $J$-th experiment using cold-starting, by Lemma C.3, a model first learns features in $\mathcal{G}$, say, in the first $s$ steps. In the $(s + 1)$-th step, cold-starting model learns a new feature since

$$
\max_{v \in \mathcal{S} \setminus \mathcal{G}} |\mathcal{T}_{1:J}| \cdot g(v; \mathcal{T}_{1:J}, \mathcal{N}_{\text{cold}}^{(J,s)}) = \max_{v \in \mathcal{S} \setminus \mathcal{G}} Jn \cdot h(v; \mathcal{G}) > \gamma,
$$

from the condition in the theorem statement. Hence, we have our conclusion for the test accuracy.

For the train time, since the following holds, we conclude $T_{\text{warm}}^{(J)} < T_{\text{cold}}^{(J)}$ when $J \geq 2$:

$$
\sum_{j \in [J]} \left| \mathcal{N}_{\text{warm}}^{(j,0)} \right| = T_{\text{warm}}^{(J)} \leq Jn < \frac{nJ(J+1)}{2} = \sum_{j \in [J]} \left| \mathcal{N}_{\text{cold}}^{(j,0)} \right| = T_{\text{cold}}^{(J)}.
$$

## C.2. Proof of Theorem 4.1

Recall that the idealized algorithm works by forgetting memorized data points while retaining previously learned features. In other words, at the initial step of the $(j+1)$-th experiment, we have $\mathcal{L}_{\text{ideal}}^{(j+1,0)} = \mathcal{L}_{\text{ideal}}^{(j)}$. Additionally, $g(v; \mathcal{T}_{1:j+1}, \mathcal{N}_{\text{ideal}}^{(j+1,0)}) = h(v; \mathcal{L}_{\text{ideal}}^{(j+1,0)})$ holds for all $v \in \mathcal{S}$ since $\mathcal{M}^{(j+1,0)} = \emptyset$.

We will show that $\mathcal{L}_{\text{cold}}^{(J)} = \mathcal{L}_{\text{ideal}}^{(J)}$ holds for all $J \geq 1$ by using induction.

When $J = 1$, by applying Lemma C.2, it holds since $\mathcal{L}_{\text{cold}}^{(1)} = \mathcal{L}_{\text{ideal}}^{(1)}$. Suppose $\mathcal{L}_{\text{cold}}^{(J-1)} = \mathcal{L}_{\text{ideal}}^{(J-1)}$ for some $J \geq 2$ and we will prove that $\mathcal{L}_{\text{cold}}^{(J)} = \mathcal{L}_{\text{ideal}}^{(J)}$. We have the following at the first step of the $J$-th experiment for all $v \in \mathcal{S}$:

$$g(v; \mathcal{T}_{1:J}, \mathcal{N}_{\text{ideal}}^{(J,0)}) = h(v; \mathcal{L}_{\text{ideal}}^{(J-1)})$$

For the cold-starting method in the $J$-th experiment, by Lemma C.2, let $s$ be the step at which the model first finishes learning features in $\mathcal{L}_{\text{cold}}^{(J-1)}$. Then, at the $(s+1)$-th step for all $v \in \mathcal{S}$:

$$g(v; \mathcal{T}_{1:J}, \mathcal{N}_{\text{cold}}^{(J,s)}) = h(v; \mathcal{L}_{\text{cold}}^{(J-1)}) \tag{3}$$

Since we assumed $h(v; \mathcal{L}_{\text{cold}}^{(J-1)}) = h(v; \mathcal{L}_{\text{ideal}}^{(J-1)})$, the cold-starting method starts to behave identically to the ideal method from the $(s+1)$-th time step onwards, by Lemma C.2, resulting in $\mathcal{L}_{\text{cold}}^{(J)} = \mathcal{L}_{\text{ideal}}^{(J)}$.

$\left| \mathcal{N}_{\text{ideal}}^{(J,0)} \right| < |\mathcal{T}_{1:J}|$ for $J \geq 1$ since $\left| \mathcal{L}_{\text{ideal}}^{(J)} \cap \mathcal{S}_c \right| \geq \tau$ for some class $c \in [C]$ due to Assumption 3.3. Thus, the training time of the idealized method, $T_{\text{ideal}}^{(J)}$, is as follows:

$$\sum_{j \in [J]} \left| \mathcal{N}_{\text{ideal}}^{(j,0)} \right| = T_{\text{ideal}}^{(J)} < T_{\text{cold}}^{(J)} = \sum_{j \in [J]} \left| \mathcal{N}_{\text{cold}}^{(j,0)} \right| = \frac{nJ(J+1)}{2}$$

# D. Omitted Algorithms

In this section, we provide detailed training algorithms for our proposed learning framework. Algorithm 2 outlines the standard training method within our learning framework. Subsequently, we compare the Cold-starting, Warm-starting, and Ideal methods using the given abstract algorithm in the following algorithms.

---

**Algorithm 2** Training Process

---

**Require:**
- $\mathcal{L}$: Set of learned features
- $\mathcal{M}$: Set of memorized data points
- $\mathcal{T}$: Training dataset
- $\gamma$: Threshold for learning features
- $\tau$: Threshold for the number of learned features a data point needs to be considered well-classified

**function** TrainingProcess($\mathcal{L}, \mathcal{M}, \mathcal{T}, \gamma, \tau$)

  **Initialize:**

    $\mathcal{N} \leftarrow \{(\boldsymbol{x}, y) \in \mathcal{T} : |\mathcal{V}(\boldsymbol{x}) \cap \mathcal{L}| < \tau \land (\boldsymbol{x}, y) \notin \mathcal{M}\}$

    $s \leftarrow 0$

  **while** $\mathcal{N} \neq \emptyset$ **do**

    $s \leftarrow s + 1$

    $g(v; \mathcal{T}, \mathcal{N}) \leftarrow \frac{1}{|\mathcal{T}|} \sum_{(\boldsymbol{x}, y) \in \mathcal{N}} \mathbb{1}(v \in \mathcal{V}(\boldsymbol{x}))$ for $v \in \mathcal{S}$

    $v_s \leftarrow \arg\max_{u \in \mathcal{S} \setminus \mathcal{L}} g(u; \mathcal{N})$   break ties arbitrarily

    **if** $g(v_s; \mathcal{T}, \mathcal{N}) \geq \gamma / |\mathcal{T}|$ **then**

      $\mathcal{L} \leftarrow \mathcal{L} \cup \{v_s\}$

      $\mathcal{N} \leftarrow \{(\boldsymbol{x}, y) \in \mathcal{N} : |\mathcal{V}(\boldsymbol{x}) \cap \mathcal{L}| < \tau\}$

    **else**

      $\mathcal{M} \leftarrow \mathcal{M} \cup \{(\boldsymbol{x}, y) \in \mathcal{N} : |\mathcal{V}(\boldsymbol{x}) \cap \mathcal{L}| < \tau\}$

      $\mathcal{N} \leftarrow \emptyset$

    **end if**

  **end while**

  **return:** $\mathcal{L}, \mathcal{M}$

**end function**

---

---

**Algorithm 3** Cold-Starting until $J$-th Experiment

---

**Require:**
- $\mathcal{T}_{1:J}$: Training dataset
- $\gamma$: Threshold for learning features
- $\tau$: Threshold for the number of learned features a data point needs to be considered well-classified

**Initialize:**

  $\mathcal{L}^{(0)} \leftarrow \emptyset$

  $\mathcal{M}^{(0)} \leftarrow \emptyset$

**for** $j = 1$ to $J$ **do**

  $\mathcal{L}^{(j)}, \mathcal{M}^{(j)} \leftarrow$ TrainingProcess($\mathcal{L}^{(j-1)}, \mathcal{M}^{(j-1)}, \mathcal{T}_{1:j}, \gamma, \tau$)

  $\mathcal{L}^{(j)} \leftarrow \emptyset$

  $\mathcal{M}^{(j)} \leftarrow \emptyset$

**end for**

**return:** $\mathcal{L}^{(j)}$

---

---

**Algorithm 4** Warm-Starting until $J$-th Experiment

---

**Require:**
- $\mathcal{T}_{1:J}$: Training dataset
- $\gamma$: Threshold for learning features
- $\tau$: Threshold for the number of learned features a data point needs to be considered well-classified

**Initialize:**
  $\mathcal{L}^{(0)} \leftarrow \emptyset$
  $\mathcal{M}^{(0)} \leftarrow \emptyset$

**for** $j = 1$ to $J$ **do**
  $\mathcal{L}^{(j)}, \mathcal{M}^{(j)} \leftarrow \text{TrainingProcess}(\mathcal{L}^{(j-1)}, \mathcal{M}^{(j-1)}, \mathcal{T}_{1:j}, \gamma, \tau)$
**end for**
**return:** $\mathcal{L}^{(j)}$

---

---

**Algorithm 5** Ideal-Starting until $J$-th Experiment

---

**Require:**
- $\mathcal{T}_{1:J}$: Training dataset
- $\gamma$: Threshold for learning features
- $\tau$: Threshold for the number of learned features a data point needs to be considered well-classified

**Initialize:**
  $\mathcal{L}^{(0)} \leftarrow \emptyset$
  $\mathcal{M}^{(0)} \leftarrow \emptyset$

**for** $j = 1$ to $J$ **do**
  $\mathcal{L}^{(j)}, \mathcal{M}^{(j)} \leftarrow \text{TrainingProcess}(\mathcal{L}^{(j-1)}, \mathcal{M}^{(j-1)}, \mathcal{T}_{1:j}, \gamma, \tau)$
  $\mathcal{M}^{(j)} \leftarrow \emptyset$
**end for**
**return:** $\mathcal{L}^{(j)}$

---

