# OpenReview forum: "DASH: Warm-Starting Neural Network Training Without Loss of Plasticity Under Stationarity"
_ICML.cc/2024/Workshop/WANT — WANT@ICML 2024 Poster_

### Official Review · Reviewer_mpvC · 2024-06-13
**Review of DASH for improving plasticity of NNs**

**Confidence:** 4

**Summary:**

The paper discusses the loss of plasticity in the context of in-distribution (stationary) data distributions and proposes a method to mitigate the issue by removing noise from learned features. The method was validated in image classification in an online learning setup and compared to Shrink & Perturb, pre-trained with the same data distribution (warm) and random initialization (cold).

Recommendation: Accept

**Strengths:**

This paper provides a practical solution for online training setup to mitigate noise memorization and its impact on model generalization.

**Weaknesses:**

- Lines 345-356 (left): Regarding shrinking the weight vector, based on the description, is the assumption that the initial weight had learned important features and not noise? How is this guaranteed?
- Lines 375-377 (left): with regards to determining accuracy on previously learned data is maintained,
- Why was the number of steps used as a metric for computing training cost instead of FLOPs?
- Clarity:
	- Line 194-left: on the discussion of training time - could use clarification
	- Line 188: In remark 3.2, how is a fixed number of feature combinations guaranteed for each experiment?
	- Line 302 right: How is the noise level established $\gamma$? And the threshold of learned features $\tau$?

**Limitations:**

The experiments were limited to classification tasks.

**Suggestions:**

**Experiments**:
- It would be interesting to see if DASH preserves its performance under more complex learning objectives e.g. Bayesian learning (SVI/MCMC) or RL (which the authors mention in their discussion).

**Cosmetic**:
- For more consistency with the existing literature, consider changing the $\mathcal{L}$ to denote the "learned features" as normally it is used to denote the loss function. Similarly, consider changing $\mathcal{N}$ as it is usually used to denote a normal distribution.
- The colors within Figure 2 are not distinguishable from each other; please try to change the pattern (circle vs. square or triangle) and different colors to make it more legible. Also, the transparent lines and faint dots are too light in color to be properly interpreted.

**Other**
- Figure cross referencing is incorrect - for example in line 260 Figure 3.2 is referenced while there's no Figure numbered as 3.2 (it's Figure 2). And in line 306 Figure 4.1 is not existent - it's figure 3. And Figure 4.2 - it's Figure 4.

---

### Official Review · Reviewer_dEcw · 2024-06-13
**A strong study on plasticity in neural networks**

**Confidence:** 3

**Summary:**

The paper provides a thorough take on maintaining plasticity in neural networks with warm restarts. The authors provide extremely thorough theoretical and empirical analyses on this problem.

**Strengths:**

1. Extremely thorough experimentation and writing.
2. Strong theoretical and empirical results.
3. Ample ablations and analyses on the problem.

**Weaknesses:**

In my opinion, there are no substantial or concrete weaknesses of the paper.

**Suggestions:**

It would have been better if the paper also studied slightly larger models and a different modality like language. However, the paper is extremely strong without this as well and should be accepted.

---

### Official Review · Reviewer_sYbB · 2024-06-14
**A novel method to mitigate plasticity loss in neural networks under stationary data distributions, but lacks sufficient mathematical rigor and reproducibility resources**

**Confidence:** 4

**Summary:**

The authors propose Direction-Aware SHrinking (DASH), a method that focuses on mitigating the loss of plasticity when warm-starting neural networks in stationary data distributions. The main contributions include of their work include identification of noise memorization as the paper highlights noise memorization as the primary cause of plasticity loss in stationary settings, which was previously thought to occur mainly in non-stationary distributions. The DASH method selectively forget memorized noise while preserving useful features, aiming to retain plasticity. The approach is validated through extensive experiments on various vision classification tasks, demonstrating improved test accuracy and training efficiency compared to existing methods. Experiments conducted on datasets such as Tiny-ImageNet, CIFAR-10, CIFAR-100, and SVHN using models like ResNet-18, VGG-16, and MLP showed that DASH outperforms traditional warm-starting and other baselines in terms of both accuracy and training efficiency.

**Strengths:**

* Identifying noise memorization as a cause of plasticity loss in stationary settings is a significant contribution.
* The idea of focusing on noise memorization in stationary distributions is interesting and somewhat novel. However, the concept of plasticity loss itself is not new.
* DASH effectively addresses the identified problem, showing improvements in both accuracy and training efficiency.
* The method is validated across multiple datasets and models, providing robust evidence of its efficacy.
* The paper is generally well-written, but some sections, particularly those describing the theoretical framework, could be clearer and more concise.

**Weaknesses:**

* The theoretical framework lacks sufficient mathematical rigor. The proofs and explanations need to be more robust and comprehensive.
* The absence of provided datasets and code hinders the reproducibility and transparency of the research.
* The analysis of why DASH works in practical settings could be more detailed. There is a need for a deeper exploration of its limitations and potential drawbacks.
* Figures and tables are useful, but some could be better explained. More detailed descriptions of the experimental setup and hyperparameters would improve clarity.
*  The methodology is sound, but the theoretical framework could be more rigorously detailed. The connection between the empirical observations and the theoretical justifications could be clearer.

**Limitations:**

* The paper is well-referenced, building appropriately on existing work. However, it could engage more critically with related literature to better situate its contributions.
* The related work section is thorough but could benefit from a more critical comparison of DASH with other methods addressing plasticity loss.
* The experiments seem reproducible, but the paper lacks explicit details on dataset availability and code. More transparency in the experimental setup and providing code repositories would enhance reproducibility.
* The contributions are valuable, but the paper does not completely revolutionize the understanding of plasticity loss. It provides an incremental improvement with a specific focus on stationary data distributions.

**Suggestions:**

* Future research should explore extending DASH to non-stationary settings and other types of data distributions. Investigating its performance on more complex and varied datasets would also be valuable.
* The paper would benefit from a deeper mathematical exploration of the conditions under which noise memorization occurs and how it interacts with different model architectures.

---

### Meta-Review · Area_Chair_3dAV · 2024-06-17

**Recommendation:** Accept (Poster)
**Confidence:** 4

**Metareview:**

All reviewers champion the acceptance of this manuscript. The AC encourages the authors to take the suggestions of Reviewer sYbB into account and consider experiments on larger models.

---

### Decision · Program_Chairs · 2024-06-17

**Decision:**

Accept (Poster)

**Comment:**

We thank the authors for their time and contribution to WANT and we are pleased to share that after the reviewing process the paper has been accepted. Congratulations! We encourage the authors to consider reviewers' feedback for the improvement of the camera-ready version. We hope to see you in person at the workshop and brainstorm on efficient training research together!